# Design and Application of a Multi-Hazard Risk Rapid Assessment Questionnaire for Hill Communities in the Indian Himalayan Region

Shivani Chouhan[1*], Mahua Mukherjee[2]

[1]Research Scholar, Centre of Excellence in Disaster Mitigation and Management, Indian Institute of Technology Roorkee, Roorkee, India

[2]Professor, Centre of Excellence in Disaster Mitigation and Management, Indian Institute of Technology Roorkee, Roorkee, India

*Corresponding Author: Shivani Chouhan (*s_chouhan@dm.iitr.ac.in)*

*Corresponding Author:**

Name: Ms. Shivani Chouhan,
**Email:** *s_chouhan@dm.iitr.ac.in,*
*Telephone: +91-9675457229 , +49-1744969778*
*Postal Address: Centre of excellence in disaster mitigation and management, IIT Roorkee, Roorkee Uttarakhand, 247667, India*

**ABSTRACT**

The Indian Himalayan Region (IHR) is prone to multiple-hazards and suffers great loss of life and damage to infrastructure and property every year. Poor engineering construction, unplanned and unregulated development, and relatively low awareness and capacity in communities for supporting disaster risk mitigation is directly and indirectly contributing to the risk and severity of disasters.

A comprehensive review of various existing survey forms for Risk assessment has found that the survey questionnaires themselves have not been designed or optimised, specifically, for hill communities. Hill communities are distinctly different from low-land communities, with distinct characteristics and susceptibility to specific hazard and risk scenarios. Previous studies have, on the whole, underrepresented the specific characteristics of hill communities, and the increasing threat of natural disasters in the IHR creates an imperative to design hill-specific questionnaires for multi-hazards risk assessment.

The main objective of this study is to design and apply a hill-specific risk assessment survey form that contains more accurate information for hill communities and hill-based infrastructure and allows for the surveys to be completed efficiently and in less time. The proposed survey form is described herein and is validated through a pilot survey at several locations in the hills of Uttarakhand, India. The survey form covers data related to vulnerability from Earthquake (Rapid Visual Screening), Flood, High Wind, Landslide, Industrial, Fire Hazard in the building, Climate Change and Non-Structural Falling Hazard. The proposed form is self-explanatory,

pictorial with easy terminologies, and is divided into various sections for better understanding
of the surveyor etc.
The application process confirmed that the survey questionnaire performed well and met
expectations in its application. The form is readily transferrable to other locations in the IHR
and could be internationalised and used throughout the Himalaya.
**Keywords:** Survey, Questionnaire Design, Multi-Hazard, Rapid Visual Screening, Himalaya

## 44  1    Introduction

The Indian Himalayas considered a significant part of the world's mountain ecosystems
(Singh, 2005). The Himalayas are geologically active, delicate, and vulnerable to both natural
and human-made processes due to their structural instability and maturity (Kala, 2014).
Numerous hazards interact at most locations, resulting in cascading or synergetic effects
(Aksha *et al.*, 2020). The Indian Himalayan Region (IHR), being prone to multiple hazards,
suffers great loss of life and damage to infrastructure and properties every year (Chouhan et
al.,2022a). Multi-hazard frequency has risen in recent decades, resulting in massive socio-
economic losses. There has been a constant rise in the number of deaths, property losses,
and damage to infrastructure and facilities (Chandel and Brar, 2010). According to UNDRR
(UNDRR, n.d.), the multi-hazard concept refers to "(1) the selection of multiple major hazards
that the country faces, and (2) the specific contexts where hazardous events may occur
simultaneously, cascadingly or cumulatively over time, and taking into account the potential
interrelated effects."
Poor engineering and construction, reckless development, human intervention, unrecognized
practices, irresponsible development initiatives, and a lack of knowledge are directly and
indirectly contributing to the risk and severity of disasters (Chouhan et al., 2022b). Many
natural disasters have become human-made phenomena as a result of the spread of
irresponsible construction practices. Such disasters have a devastating socio-economic
impact on the country's economy, putting even more strain on an already stressed economy
(Disasters, 2007).
Various research work, disaster risk assessment studies and, implementation projects are
being executed by national and international organizations for disaster risk reduction in the
Himalayas. The data collection for any risk assessment in this difficult terrain is a crucial task,
as correct information documentation has played a significant role that directly or indirectly
lead to an influence in correct assessment of the risk factor (Chouhan et al.2022b).
Surveys using a well-crafted questionnaire is a proven method in the research fraternity.
Questionnaires are the backbone of every survey when it comes to data collection. Using data,
one can gain a detailed understanding of a community's hazard profile, vulnerability
interactions and their contribution to risk reduction (Buck and Summers, 2020). The survey
information is required to be coherent for data analysis since they lead to critical decisions at
many levels, represent the site's vital characters and society's expectations and requirements
too. All of these outcomes hinge, of course, on the creation of a robust site-specific survey
form. A well designed and executed Multi-Hazard Risk Assessment (MHRA) can lead to more
robust strategies for disaster risk reduction (Kala, 2014; Sekhri et al., 2020) and can facilitate
by prioritizing development planning decisions.
After studying existing survey forms and practical field survey at various locations in the Indian
Himalayas, authors found that the existing MHRA survey forms used in India have some
lacuna from the hills point of views as Himalayas have different geography, cultural,
development practices, hazard profile etc. (Chouhan et. al., 2022b). A close evaluation of the
existing survey questionnaires reveals that there is a need for IHR-specific survey
questionnaire form to facilitate a MHRA, which should be easy to understand, pictorial, and
that creates a two-way disaster sensitization of giving and getting information from the
community.
In this research paper, the journey to design and application of the proposed Hill specific
MHRA survey form has been described. The pilot survey using the proposed survey form has
been conducted at 10 schools in Uttarakhand state of India and its results identify various risk
indicators in individual building as well as the school campus.
**2   Background**
*2.1   Defining the Indian Himalayan Region*
The Indian Himalayan Region (IHR) straddles the northern latitudes of 26 20′ and 35 40′, and
the eastern latitudes of 74 50′ and 95 40′ (Sekhri et al., 2020). In India, it comprises 16.2 % of
all the geographical land and is home to 76 million people. Natural resources, biodiversity, and
ethnic variety are abundant in IHR. (Goodrich et al., 2019; Sekhri et al., 2020). It stretches
from the Indus River to the Brahmaputra River in the east. (Srivastava et al., 2015). There are
a total of 11 Indian Himalayan states and 2 Union territories as shown in Fig. 1, which have
109 administrative districts (Kala, 2014). The region is socially and economically
underprivileged, with 171 schedule tribes accounting for almost 30 % of India's total tribal
population and a high literacy rate of 79 percent. The population is growing exponentially,
putting a strain on the region's resources (COI, 2011). Tourism is a lucrative business in IHR
(NITI Aayog, 2018) and it contributes to support a lot of construction projects like hotels,
restaurants, road construction etc. across the region (Kala, 2014). Agriculture is a profitable
venture for Himalayan people, and it is mainly rain-nourished. Furthermore, climate change is
hazardous to the region's progress and hinders socio-economic development (Sekhri *et al.*,
108 2020).

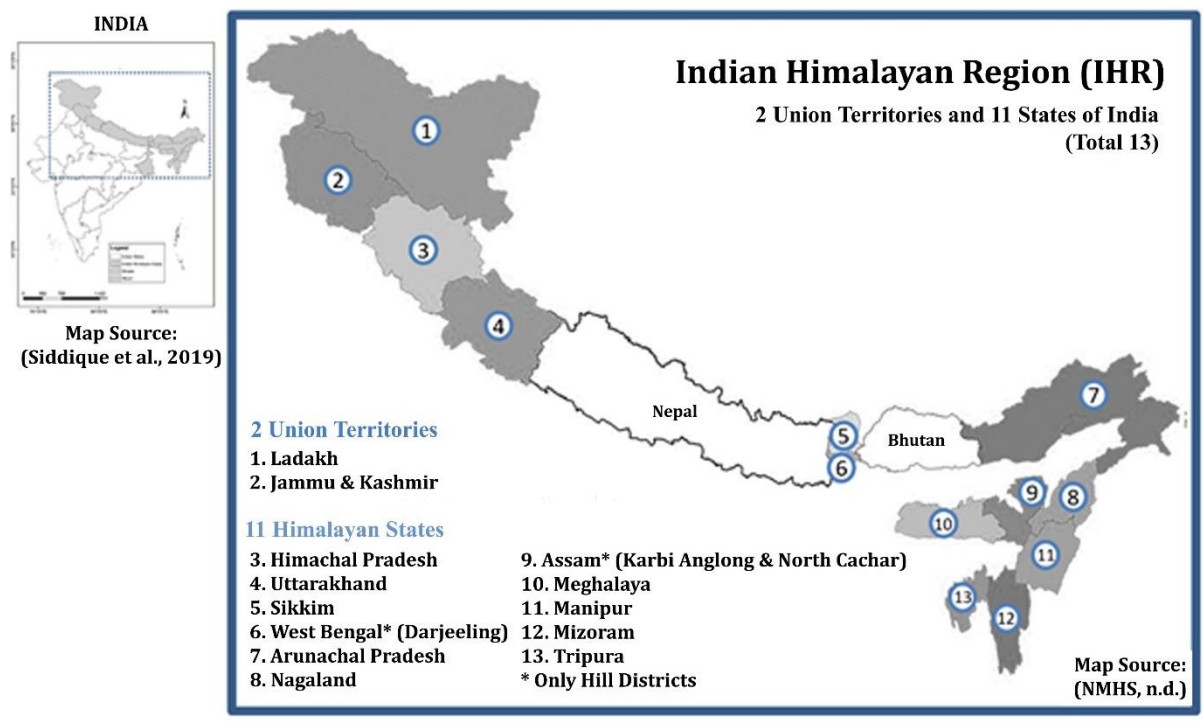

*Figure 1: Study Area: Indian Himalayan Region, Source: adapted from (NMHS, n.d.)( Siddique et. al., 2019)*
The IHR represents a significant role in the world's mountain ecosystems (Singh, 2005). IHR
attracts tourists worldwide because of its natural richness, unique biodiversity, and cultural
diversity (NITI Aayog, 2018,; Gaur and Kutro, 2018). The number of pilgrims has risen
dramatically in prominent pilgrim centres across the Himalayas over the ages (Kala, 2014),
putting extra stress on these resources and posing a danger of socioeconomic loss.
*2.2  Multi Hazards in IHR*
Being geologically young and expanding (Wester et al., 2019), the IHR is vulnerable to natural
disasters (Gautam et. al., 2013). The Himalaya, the world's highest mountain range is
geologically active, fragile, and susceptible to natural and man-made processes (Kala, 2014).
Indian geography, climate, topography, and population growth all contribute to its high risk and
vulnerability (Sharma *et al.*, 2017). Mountain hazards are widespread, and hills characteristics
of fragility, restricted accessibility, marginality, and heterogeneity (Gerlitz et al., 2016) may
turn a hazard into a catastrophe, transforming mountains into high-risk zones. Furthermore,
mountains need a long time to recover from disruptions  (Sekhri *et al.*, 2020).
Multi-Hazard Frequency has risen in recent decades, resulting in massive socio-economic
losses (Rehman et al., 2022). Unrecognized practices, irresponsible development initiatives,
and a lack of knowledge contribute to disasters having a more significant effect. One of the
most challenging aspects of natural hazards risk assessment is determining how to estimate
the risk of several hazards in the same region and how they interact (Hackl et. at., 2015).
In the recent decade, severe earthquakes, floods, and landslides have devastated IHR,
including the M 7.6 Kashmir earthquake in 2005, the Malpa Landslide in 2009, the M 6.8
Sikkim earthquake in 2011, the 2013 Uttarakhand flash flood, and others, affecting
approximately thousands of deaths and property losses (MHA, 2011; BMTPC, 2019; Kumar
et al., 2016). Table 1 illustrate and describe the major hazard events that have occurred
historically in the Indian Himalayan region.
*Table 1: Major Disaster Events in IHR, Source: adapted from (BMTPC, 2019; Kumar et al., 2016).*

| SN | Date | Location (Latitude, Longitude) | Place | Indian Himalayan State | Hazard/ Magnitude | Casualties | Source |
|---|---|---|---|---|---|---|---|
| 1 | 1869, Jan 10 | (25.00, 93.00) | Near Cachar | Assam | Earthquake 7.5 Mw | Unknown | Kumar et al., 2016 |
| 2 | 1885 May 30 | (34.10, 74.60) | Sopor | Jammu & Kashmir | Earthquake 7.0 Mw | Unknown | Kumar et al., 2017 |
| 3 | 1897 Jun 12 | (26.00, 91.00) | Shillong plateau | Meghalaya | Earthquake 8.7 Mw | 1500 | Kumar et al., 2018 |
| 4 | 1905 Apr 04 | (32.30, 76.30) | Kangra | Himachal Pradesh | Earthquake 8.0 Mw | 19,000 | Kumar et al., 2019 |
| 5 | 1918 Jul 08 | (24.50, 91.00) | Srimangal | Assam | Earthquake 7.6 Mw | Unknown | Kumar et al., 2020 |
| 6 | 1930 Jul 02 | (25.80, 90.20) | Dhubri | Assam | Earthquake 7.1 Mw | Unknown | Kumar et al., 2021 |
| 7 | 1943 Oct 23 | (26.80, 94.00) | Assam | Assam | Earthquake 7.2 Mw | Unknown | Kumar et al., 2022 |
| 8 | 1950 Aug 15 | (28.50, 96.70) | Arunachal Pradesh–China Border | Arunachal Pradesh | Earthquake 8.5 Mw | 1526 | Kumar et al., 2023 |
| 9 | 1975 Jan 19 | (32.38, 78.49) | Kinnaur | Himachal Pradesh | Earthquake 6.2 Mw | Unknown | Kumar et al., 2024 |
| 10 | 1988 Aug 06 | (25.13, 95.15) | Manipur–Myanmar border | Manipur | Earthquake 6.6 Mw | 1000 | Kumar et al., 2025 |
| 11 | 1991 Oct 20 | (30.75, 78.86) | Uttarkashi, UP | Uttarakhand (now) | Earthquake 6.6 Mw | 2000 | Kumar et al., 2026 |
| 12 | 1998 Aug 18 | (30.01, 80.04) | Malpa, Pithoragarh district | Uttarakhand (now) | Landslide | 380 | Kumar et al., 2027 |
| 13 | 1999 Mar 29th | (30.41, 79.42) | Chamoli District, UP | Uttarakhand (now) | Earthquake 6.8 Mw | 100 | Kumar et al., 2028 |
| 14 | 2005 Oct 08th | (34.48, 73.61) | Kashmir | Jammu & Kashmir | Earthquake 7.6 Mw | 74,500 | Kumar et al., 2029 |
| 15 | 2006 Feb 14th | (27.37, 88.36) | Sikkim | Sikkim | Earthquake 5.7 Mw | 0 | BMTPC, 2019 |
| 16 | 2010 Aug 06th | (34.15, 77.57) | Leh | Ladakh (now) | Cloudburst | 257 | BMTPC, 2019 |
| 17 | 2011 Sep 18th | (27.7, 88.2) | Sikkim Nepal border | Sikkim | Earthquake 6.8 Mw | 60 | Kumar et al., 2016 |
| 18 | 2012 July-Aug | (26.20, 92.93) | Assam | Assam | Floods | 91 | BMTPC, 2019 |

| 19 | 2012 Aug-Sep | (30.72, 78.43), (30.28, 78.98), (29.84, 79.76) | Uttarkashi, Rudraprayag & Bageshwar | Uttarakhand | Floods | 52 | BMTPC, 2019 |
|----|----|----|----|----|----|----|----|
| 20 | 2013 June 16th | (30.06, 79.01) | Uttaranchal | Uttarakhand (now) | Flood, Landslide, Cloud Burst | 5748 | Kumar et al., 2016 |
| 21 | 2014 Sep | (33.27, 75.34) | Jammu & Kashmir | Jammu & Kashmir | Flood, Cloud Burst | 277 | Kumar et al., 2016 |
| 22 | 2016 Jan 04th | (24.81, 93.93) | Imphal, Manipur | Manipur | Earthquake 6.7 Mw | 8 | BMTPC, 2019 |


The Himalayan region is among the most seismically active in the world due to the collision of
the Indian and Eurasian plates. A series of four major earthquakes has occurred within a short
span of 53 years (Srivastava et al., 2015); namely Shillong (1897), Kangra (1905), Bihar-Nepal
(1934) and Assam-Tibet (1950). Tectonic activities in the mountains constantly threaten the
stability of the mountains, being an active region. One of the most frequent natural disasters
in the Himalayas occurs when large landslides occur, destroying infrastructures, destroying
trees, and killing people. Landslides cause huge social and economic losses to mountain-
dwelling populations.(Sarkar et al., 2015). The areas which are close to the River valley has
witnessed a large number of mass movements during recent years (Srivastava et al., 2010).
A recent flash flood, along with a debris flow at Kedarnath on 16-17 June 2013, which claimed
over a thousand lives, was caused by cloudbursts and landslides breaching temporary dams
along river valleys (Allen, 2015). More than 82 percent of the world's population lived on land
affected by floods between 1985 and 2003 (Mouri *et al.*, 2013). There is an increase in forest
fire frequency globally, especially in Asia. There are major environmental and ecological
impacts caused by wildfires, which can result in the fatalities of tens of thousands of people
and massive property losses (Parajuli et al., 2020).
*2.3  Need of Study*
Without a comprehensive evaluation of multi-hazards, it is impossible to develop any concrete
policy measures to combat the potential risk posed by multiple hazards.(Sekhri et al., 2020)
IHR being prone to Multi Hazards (Kala, 2014), Risk Resilient Development planning is the
only way to prepare Himalayan community from upcoming disasters.
It is well known that the Himalayas are a high-risk area for multi-hazards (Pathak et al., 2019),
although fewer risk assessments have been conducted in the IHR region. An assessment of
hazards generally focuses on a single threat, such as landslides, earthquakes, or flooding. As
a result, physical processes are considered in isolation. In most areas of the Himalayas,
hazards are interrelated and generate cascading effects or synergies which make the entire
region vulnerable (Sekhri et al., 2020). Probabilistic risk frameworks have been proposed, but
as a result of a lack of quality and quantity of data, these approaches are seldom feasible in
developing countries (Sanam et al., 2020). Furthermore, the existing risk assessment
models/tools for a specific hazard in the region has limited application and effectiveness from
a policy standpoint (Sekhri et al., 2020).
Researchers are involved in a number of research projects in IHR in the field of assessing the
risk of disasters in India, though there have been very few assessments of hazards associated
with the IHR region, none of which incorporate multi-hazards (Wester et al., 2019) In addition,
risk resulting from a single hazard is not applicable and cannot be considered effectively in
policy analysis in the region (Sekhri et al., 2020).
The comparative study of some of the most used survey forms to assess risk in India is shown
in the Table 2. Every survey form has its own unique features. In some cases, the focus is
largely on one particular hazard and the other hazards are minor. The detail of all the
mentioned survey forms will be explained later in Table 4 in this paper. It has been observed
from the Table 2 that none of the forms (SN 1 to 6) are focusing on Multi Hazard Risk
calculation/identification as per IHR Scenarios, which is not only prone to earthquakes, but
also prone to floods, landslides, high winds, industrial hazards and at building level falling
hazard (Non-Structural Hazard), fire and electrical hazards etc.
*Table 2: Comparison between survey forms used in India to assess Risk*

| SN | | 1 | 2 | 3 | 4 | 5 | 6 | 7 |
|---|---|---|---|---|---|---|---|---|
| Developed by/for | | ARYA | FEMA | NDMA | IIT-B | HPSDMA | BMTPC | MH-RVS (Proposed) |
| Source: adapted from | | Arya, 2006 | FEMA, 2015 | NDMA, 2020 | Sinha & Goyal, 2001 | Kumar et al., 2016 | BMTPC, 2019 | Author |
| Understanding | Pictorial | | | | | ✓ | | ✓ |
| IHR is prone to Multi Hazard | Earthquake | ✓ | ✓ | ✓ | ✓ | ✓ | ✓ | ✓ |
| | Flood | | | ✓ | | ✓ | ✓ | ✓ |
| | High Wind | | | | | | ✓ | ✓ |
| | Landslide | ✓ | ✓ | ✓ | | ✓ | ✓ | ✓ |
| | Fire and Electrical | | | | | ✓ | | ✓ |
| | Industrial | | | | | | | ✓ |
| | Climate Change | | | | | | | ✓ |
| | Non-Structural /Falling Hazard | ✓ | ✓ | ✓ | ✓ | ✓ | | ✓ |


Furthermore, while working with data collection teams on the ground during DRR Projects, the
authors have observed that surveyors face several problems, such as the technically
advanced language of the existing survey form, which requires trained technical personnel to
fill out, and this leads to costly human resources. Secondly, no graphical explanation of the

form leads to understanding, which further leads to incorrect data collection. Thirdly, Surveyors are not able to convey correct objective to the respondent, creates no interest to response to reply further. Fourthly, most of the above-mentioned forms are not hill specific. MHRA survey forms need to be made easy, simple, informative, with simple language or/and visual explanation, for surveyors as well as respondents to get connected to it for giving and receiving information.

Indian Himalayan Region is also the point of attraction for tourists and pilgrims globally, and tourism plays an imperative role in enhancing the economy of the Himalayan state. Thus, safety is the immense need of the government at various levels.

There is no such survey form for comprehensive database for the IHR Region for informed decision-making, related to multi hazard and other aspects of sustainable hill development. Considering the IHR scenarios, there is immense need for a Hill specific survey form, that can help to gather important information from the field and help in Risk assessment for further decision making, to prepare the hill community from future disasters.

## 3   Multi Hazard Survey Framework

### 3.1   Survey Form design methodology

The survey methodologies start with a few recommendations for designing a good survey, like (1) the survey form should satisfy the objectives of the research, (2) there should appropriate (but not very long) length of questionnaires coving all essential parts, (3) questions should convey a single thought at a time, (4) language should be simple and easy to understand by the surveyors as well as the respondent, (5) multiple choice questions are mostly preferred to increase response rate, reduce time and patterned the responses, (6) The survey should be concrete and conform to the respondent's perspective, (7) the use of unclear words should be avoided (8) it should meet the survey logic i.e. there is no further progress or possibility of further correspondence from the respondent, if the logic is flawed.  It takes practice and verification to ensure that when considering an option only the next logical question comes to mind (Roopa and Rani, 2012).

### 3.2   Methodology Adopted

To gather beneficial and appropriate information related to multi-hazards in the Himalayan region, careful attention must be given to the design of the questionnaire that covers all the important contributing factors from various identified hazards and fulfils all the gaps identified from the existing survey form and field experience. Designing an effective questionnaire, it takes time, effort, and a variety of stages. The methodology to prepare the Multi-Hazard Survey form for Indian Himalayan Region is shown in Fig. 2.

A number of Disaster Risk Reduction projects conducted in Indian Himalayan Region provided
Author 1 with a rare opportunity to be part of a Data Collection team. As a result of these
projects, author has been able to interact on the ground with hill communities and surveyors
and learned that there are several gaps in the existing survey forms (Section 3.4) from both a
Himalayan and surveyor perspective. MHRA Survey form contains all the gist of data collection
experience. This research paper is based on a comprehensive literature review (Section 3.3)
as well as field experience.
To ensure that the survey form was designed in accordance with Disaster Risk Assessment
requirements, Hill specific hazards, important components, question sequence and layout,
simple language, disaster sensitization, and two-way information sharing (giving and
receiving), some initial considerations were taken into account.
We have designed a draft MHRA survey form (Section 4.1) and applied it to some of the
buildings in five villages in Uttarakhand (Fig. 5). An initial pilot survey has been conducted at
10 schools (section 4.2) using the proposed survey form with content and graphical inputs.
The results and observations relating to the Pilot survey are discussed in sections 4.2 and 4.5
of this paper.

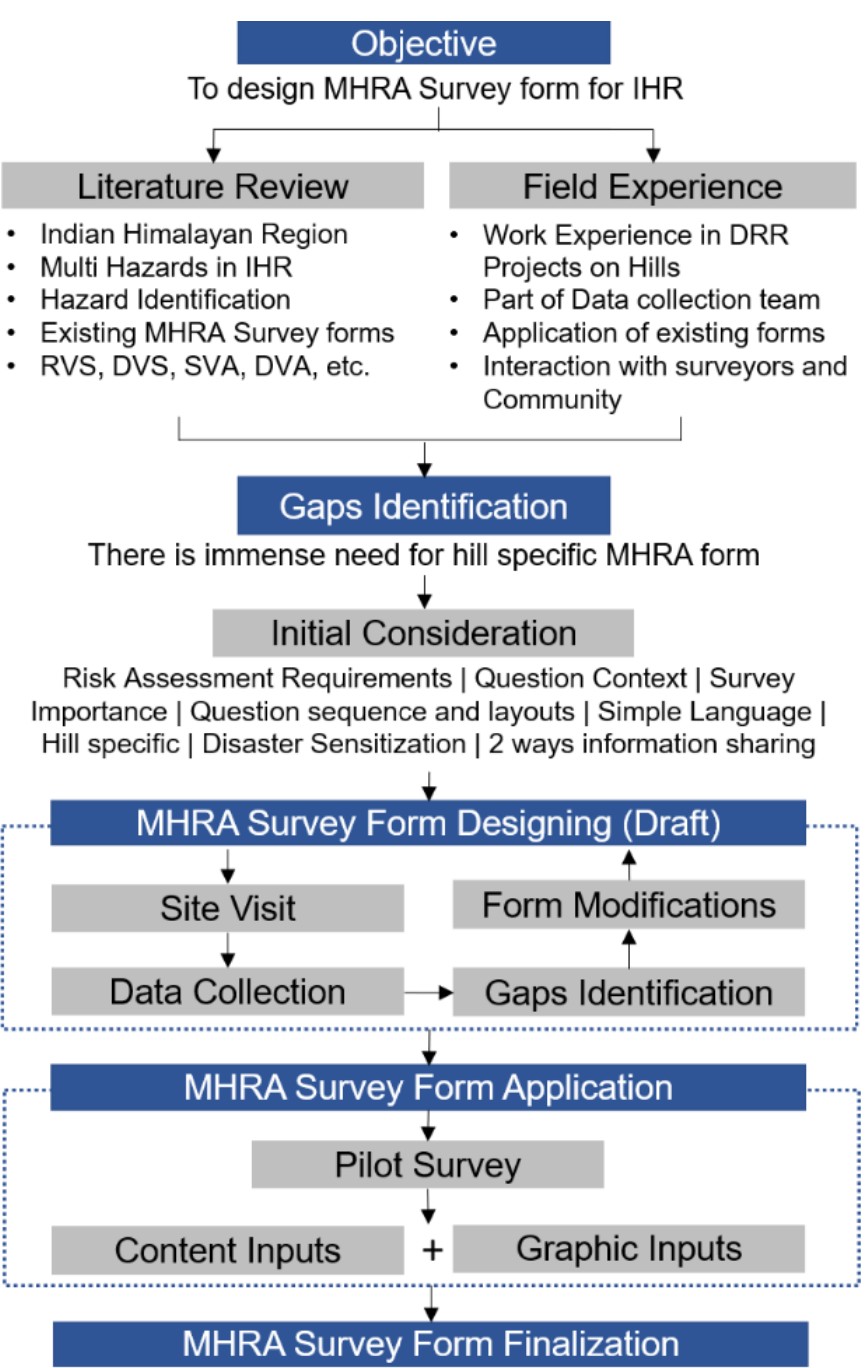


240                              *Figure 2: Methodology adopted by author*

*3.3    Existing Multi Hazard Risk Assessment (MHRA) Survey Forms*
The spread of non-engineering construction, unrecognized construction and planning
practices, reckless developmental activities, and a lack of awareness increase the impact of
disasters. IHR being seismically active, as shown in the seismic zonation map of India, creates
the importance of Risk assessment of existing buildings. Earthquakes are feared because
they are so unpredictable. Yet, as we often hear, "Earthquakes don't kill, Buildings do"
(attributed to Francesca Valli, Change Management Thought-Leader), and as the detailed

assessment is limited by the number of homes and the cost, one of the considering approaches is Rapid Visual Screening (RVS) that is used for seismic vulnerability assessment. Using this methodology, a risk assessment has been conducted for areas subjected to earthquakes (Kumar et al., 2016).

### 3.3.1 Seismic Zonation Map of India

The first seismic zoning map of India was published in 1935 by the Geological Survey of India (G. S. I.) (Fig. 3) (A. K. Mohapatra, 2010). Based on the damage earthquakes caused in various parts of India, this map has undergone numerous modifications (IS-code1893-1, 2002) (Marcussen, 2017), (Khattri *et al.*, 1984) since its original creation As per the Seismic zonation map, India is divided into four distinct seismic risk zones shown here by colour (Dunbar, 2003) in Fig. 3 below:

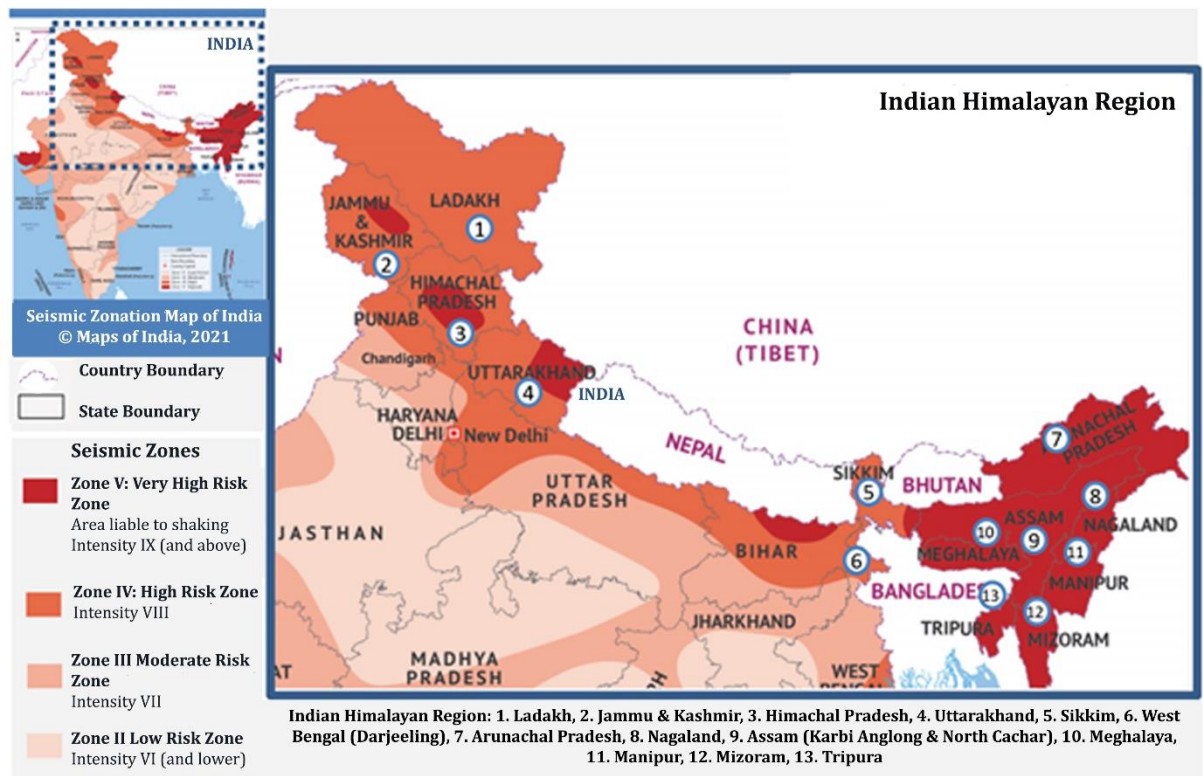

*Figure 3: Seismic Zonation Map of India, Source: adapted from (pp. Map of India, 2021)*

### 3.3.2 About RVS

Applied Technology Council (ATC) developed the RVS method in the late 1980s and published it in the FEMA: 154 in 1988. In later versions, it was revised in FEMA: 178-1989, 1992 (revised), FEMA: 310-1998, and FEMA: 154-1988, 2002 (revised), for rapid visual screening of buildings. (Kumar et al., 2016)

Rapid Visual Screening (RVS) avoids the need for structural calculations by using a visual method. An evaluator determines damageability grade by identifying (a) the primary structural

lateral load resisting system as well as (b) the structural features of the building that can impact
seismic performance in combination with that system. The process of inspecting, gathering
data, and deciding on the next course of action occurs on site and may last several hours,
depending on the size of the building (Arya, 2006; Arya, 2006b).
*3.3.2.1   Uses of RVS Results:*
The foremost uses of this technique concerning seismic advancement of existing buildings are
to assess a building's seismic vulnerability to categorize it further. It is used to determine the
structural vulnerability (damageability) of buildings and determine the seismic rehabilitation
requirements. In cases where further assessments are not considered necessary or are not
feasible, retrofitting requirements are simplified (to a collapse prevention level) (Arya, 2006a;
Arya, 2006b).
**3.3.3   Multi Hazard Risk Assessment used in India**
*3.3.3.1   RVS Methodology Proposed by Prof. Anand S Arya for Masonry Buildings*
This RVS procedure that was designed for the Indian context follows a grading system where
the screener identifies the primary load-resisting system of the building and determines
parameters that may be modified to improve seismic performance of the structure (NDMA,
284     2020)

Rapid Visual Screening form of Masonry Buildings developed by Prof. Anand S Arya consist
of zoning, according to Indian conditions, and buildings with importance are given
consideration. Also, special hazards (liquefiable area, landslide prone area, plan irregularities,
and vertical irregularities) and falling hazards are taken into account. Finally, a grading system
was performed in the buildings. Refer (Arya, 2006a) for detailed RVS survey forms for
masonry buildings.
*3.3.3.2   RVS Methodology Proposed by Prof. Anand S Arya for RC frame or Steel Frame*
The Rapid Visual Screening form of Reinforced Concrete frame and Steel Frame for Seismic
Hazards developed by Prof. Anand S Arya  has 6 components (i) general information (ii)
Building typology based on foundation type, roof, floor, etc. (iii) Structural frame type (iv)
Special Hazard (v) Non-Structural building components (vi) Damageable Grades (Arya,
2006b).
Seismic safety features of RC Frame Buildings consist of parameters like Frame Action,
Presence of Soft Storey, Short Column Effect, Concept of Weak Beam Strong Column,
Pounding of Buildings, Building Distress and Other important features, Water Seepage,
Corrosion of Reinforcement, Quality of Construction, Quality of Concrete and non-structural
falling hazards. Refer (Arya, 2006a; Arya, 2006b) for detailed RVS Survey form for RC and
steel buildings.

### 3.3.3.3 RVS Procedure developed by Dr. Sudhir K Jain

In this method, a checklist for pre-screened buildings is prepared based on Indian conditions.
It is one of the first methodologies in India featuring a points system. Performance scores are
calculated based on factors such as zone, architectural considerations, structural parameters,
and geotechnical characteristics. In India, this method is used in many locations, with the first
applications being in Gujarat after the Bhuj earthquake (Jain et al., 2010).

### 3.3.3.4 RVS form developed by NDMA 2020

In the Disaster Management Act of 2005, a paradigm shift from Relief-centric approach to
Mitigation- and Preparedness-centric approach is sought, with continued emphasis on
proactive, holistic and integrated Response. With this Act in mind, NDMA initiated a series of
discrete, comprehensive, and integrated initiatives. Among the recommended actions was
assessing earthquake risk within the existing built environment.
NDMA developed this report to make end users aware of RVS's outcomes by presenting RVS
in clear and tangible terms. On the basis of discussions with the relevant domain experts,
NDMA have developed recommended forms for Pre-Earthquake and Post-Earthquake Level
1 Assessments of 7 building typologies (i. Reinforced Concrete Building, ii. Burnt Clay Bricks
Building, iii. Confined Masonry Building, iv. Random Rubble Masonry Building, v. Mud House,
vi. Dhajji Dewari, vii. Ekra House). A form is developed to categorize the different building
attributes into three categories: Red (High Risk), Yellow (Moderate Risk), and Green (Low
Risk) (NDMA, 2020).

### 3.3.3.5 Seismic Vulnerability Assessment by Prof. Ravi Sinha and Prof. Alok Goyal

Prof. Ravi Sinha and Prof. Alok Goyal from Indian Institute of Technology Bombay (IIT-B)
prepared a "National Policy for Seismic Vulnerability Assessment of Buildings and Procedure
for Rapid Visual Screening of Buildings for Potential Seismic Vulnerability". A key feature of
this procedure is that it allows a trained evaluator to conduct a walkthrough of the building to
determine vulnerability. It is compatible with GIS-based city databases, and can also be used
for a variety of other planning and mitigation tasks (Sinha and Goyal, 2001).
RVS analysed 10 different types of building, based on the materials and construction types
most commonly found in urban areas. There were both engineered and non-engineered
constructions (built according to specifications) in this category (Sinha and Goyal, 2001).
*3.3.3.6 Building Vulnerability form developed by HPSDMA & TARU*
A form originally prepared by TARU consultancy and the Himachal Pradesh State Disaster
Management Authority (HPSDMA) is shown in (Kumar et al., 2016). A building is visually
examined by an experienced screener as part of RVS to identify features that contribute to
seismic performance. This method is known as a 'sidewalk survey.' In this side walk survey,
checklists are provided for each of the five types of buildings i.e., RC frames, brick masonry,
stone masonry, Rammed Earth, and hybrid (Kumar et al., 2016).
*3.3.3.7 Vulnerability Atlas of India developed by BMTPC*
Building Materials and Technology Promotion Council (BMTPC) published the Vulnerability
Atlas of India as its first edition in 1997 (BMTPC, 2019). It was hailed as "useful tool for policy
planning on natural disaster prevention and preparedness, especially for housing and related
infrastructures". First of its kind, it provided a means for assessing not only district-level
hazards, but also the vulnerability and risks of housing stock. It was greatly utilized by State
Governments and their agencies in order to develop micro-level action plans on how to reduce
the impact of natural disasters since buildings and housing are commonly damaged or
destroyed due to natural disasters, resulting in life losses and disruptions to socio-economic
activities.
The revised Atlas 2019 reflects advances in scientific & technical knowledge, addition of new
datasets, results of disasters caused by earthquakes and cyclones, possible damage from
landslides, floods, thunderstorms, failures of roads and trains during disasters, changes in the
political map of the country, and new statistics on walling and roofing data of houses (BMTPC,
2019). Table 3 and Fig. 4 shows different Housing typologies used in BMTPC, based on wall
and roof type and material identified in India and also their Damage risk under various hazard
intensities.
*Table 3: Damage Risk to various Housing Category identified by BMTPC under various Hazard Intensities*
*(BMTPC, 2019)*

| Category (Type of Wall and Roof) | Earthquake Intensity MSK | | | | Wind Velocity m/s | | | | Flood Prone |
|---|---|---|---|---|---|---|---|---|---|
| | ≥ IX | VIII | VII | ≤ VI | 55 & 50 | 47 | 44 & 39 | 33 | |
| A1. Mud wall (All roofs) | VH | H | M | L | VH | H | M | L | VH |
| A2.a. Unburned Brick Wall (Sloping roofs) | VH | H | M | L | VH | H | M | L | VH |
| A2.b. Unburned Brick Wall (Flat roofs) | VH | H | M | L | VH | H | M | L | VH |
| A3.a. Stone Wall (Sloping roofs) | VH | H | M | L | VH | H | M | L | VH |
| A3.b. Stone Wall (Flat roofs) | VH | H | M | L | H | M | L | L | VH |
| B.a. Burned Brick Wall (Sloping roofs) | H | M | L | VL | H | M | M | L | H |
| B.b. Burned Brick Wall (Flat roofs) | H | M | L | VL | M | L | L | VL | H |
| C1.a. Concrete Wall (Sloping roofs) | M | L | VL | NIL | H | M | M | L | L |
| C1.b. Concrete Wall (Flat roofs) | M | L | VL | NIL | L | VL | VL | VL | L |

| | | | | | | | | | |
|---|---|---|---|---|---|---|---|---|---|
| C2. Wood Wall (All roofs) | M | L | VL | NIL | VH | H | M | L | H |
| C3. Ekra Wall (all roofs) | M | L | VL | NIL | VH | H | M | L | H |
| X1. GI (Galvanised Iron) and other metal sheets (All roofs) | M | VL | NIL | NIL | VH | H | M | L | H |
| X2. Bamboo, Thatch, Grass, Leaves etc. (All roofs) | M | VL | NIL | NIL | VH | VH | H | L | VH |
| VH: Very High Risk; H: High Risk; M: Moderate Risk; L: Low Risk; VL: Very Low Risk | | | | | | | | | |


**Housing Category : Wall Types**
**Category - A** : Buildings in field-stone, rural structures, unburnt brick houses, clay houses
**Category - B** : Ordinary brick building; buildings of the large block & prefabricated type, half-timbered structures, building in natural hewn stone
**Category - C** : Reinforced building, well built wooden structures
**Category - X** : Other materials not covered in A,B,C. These are generally light.
***Notes*** : 1. Flood prone area includes that protected area which may have more severe damage under failure of protection works. In some other areas the local damage may be severe under heavy rains and chocked drainage.
2. Damage Risk for wall types is indicated assuming heavy flat roof in categories A, B and C (Reinforced Concrete) building
3. Source of Housing Data : Census of Housing, GOI, 2011

**Housing Category : Roof Type**
**Category - R1** - Light Weight (Grass, Thatch, Bamboo, Wood, Mud, Plastic, Polythene, GI Metal, Asbestos Sheets, Other Materials)
**Category - R2** - Heavy Weight (Tiles, Stone/Slate)
**Category - R3** - Flat Roof (Brick, Concrete)
EQ Zone V : Very High Damage Risk Zone (MSK > IX)
EQ Zone IV : High Damage Risk Zone (MSK VIII)
EQ Zone III : Moderate Damage Risk Zone (MSK VII)
EQ Zone II : Low Damage Risk Zone (MSK < VI)
*Level of Risk : VH = Very High; H = High;*
*M = Moderate; L = Low; VL = Very Low*
* Total No.of Houses excluding Vacant/Locked Houses

bmtpc **Building Materials & Technology Promotion Council**        Peer Group, MoHUA, GOI


*Figure 4: Damage Risk and Housing category identified by BMTPC (BMTPC, 2019)*
### 3.3.4   Multi Hazard Risk Assessment used Globally
#### 3.3.4.1   FEMA 154
The FEMA handbook demonstrates how to rapidly identify, inventory, and rank buildings that
are at high risk of causing death, injury, or severe damage in the event of an earthquake.
Rapid Visual Screening (RVS) can be carried out with a short exterior inspection, lasting 15 to
30 minutes, by trained personnel using the data collection form in the handbook. The guide is
targeted at building officials, engineers, architects, building owners, emergency managers,
and citizens who are interested in the topics.
Its purpose was to provide an evaluation of the seismic safety of a large inventory of buildings
quickly and inexpensively, with minimal access to the buildings, and to identify those that
require more detailed examination. FEMA 154 was developed by ATC under contract to FEMA
(ATC-21 Project) in 1988. As with its predecessors, the Third Edition aims to identify,
inventory, and screen buildings that present a potential risk. This latest version includes major
improvements, such as: updating the Data Collection Form and including an optional more
detailed page, preparing additional reference guides, and including additional building types
that are common, considerations such as existing retrofits, additions to existing buildings, and
adjacency, and many others (FEMA, 2015).
*3.3.4.2  Flood Vulnerability Assessment survey*
The Flood Vulnerability Assessment survey form prepared by the Asian Institute of Technology
(AIT) Bangkok and Climate Technology Centre and Network (CTCN) (Peiris, 2015) has 5
Sections: (i) General Information (ii) Type of Building (iii) Flood damage and cost (iv) Flood
emergency response (v) Effect on livelihood and income and was designed for Residential,
Institutional, Commercial/Industrial damages and Infrastructure damages. Refer (Singh, 2005)
for detailed Survey form.
*3.3.4.3  Landslide Vulnerability Assessment survey*
Scientists and researchers focus more on researching landslide susceptibility and the hazard
component rather than assessing the vulnerability of buildings to landslides. Even when the
same construction material is used, construction practices vary across the country. Currently,
there is no standard method for determining building vulnerability by using indicators.
The parts covered by Landslide risk assessment survey forms are (i) General information (ii)
Building Function (iii) Vulnerability Indicators like Architectural Features, Material
Characteristics, Structural Features, Geographical features, and quality of Workmanship,
Construction & maintenance, etc. which are also covered during RVS and has been covered
in the proposed survey form CitSci, GIS based data collection app for landslide (Singh et al.,

396    2019).

*3.4   Features required for a Multi Hazard Survey Form for IHR*
**3.4.1   Gaps Identified in existing survey forms**
Existing Survey forms have their strengths & weaknesses. After studying various survey forms
for Risk assessment prepared by various national and international authorities, it is observed
that hill-specific survey forms that can take care of multiple aspects of risk and sustainability
assessment together do not exist. Available forms are complicated, not-so user friendly,
consisting of terminologies difficult to communicate and comprehend, no pictorial clues for
understanding, involve several rounds of calculations for coherent multi-hazard risk evaluation
using the data, and most importantly, they are not hill site-specific or designed for the Indian
Himalayan region.
Hills have their own situation, condition, geography, climate, development trends, construction
practices, culture, etc., and they are distinctly different from other regions. RVS is mostly used
in India to assess the visual structural vulnerability of the building, as it involves no structural
calculations. On the other hand, SVA (Simplified Vulnerability Assessment) and DVA (Detailed
Vulnerability Assessment) are for the detailed structural survey of a building, and therefore
more precise and use engineering information along with more explicit data on ground motion.

 Data filling is not easy enough for the surveyor and requires a very high level of engineering
knowledge, skills, and experience. Pictorial explanation from surveyor point of view can ease
the communication. Most of the survey forms are focused on single hazard, (mostly for seismic
evaluation of a building) irrelevant of multi hazard from Himalayan point of view, and how
prone a building's location is to other hazards. Integration between risk understanding and
sustainable development is too limited or non-existent. Thus, it has been observed that there
is an immense need to design hill-specific questionnaires for multi-hazards risk assessment
for Indian Himalayan Region.
**3.4.2 Comparative Study of some risk assessment survey forms mostly used in India**
Table 4 shows the comparative analysis of Risk assessment survey forms developed by
various organizations and mostly used in India with the proposed Multi-Hazard RVS. Forms
have been compared on various sections like typology, General Information, History of
Disasters, Site Conditions, Building geometry, structural and non-structural component of a
building etc.
*Table 4: Comparative Study of some risk assessment survey forms mostly used in India*

| | | 1 | 2 | 3 | 4 | 5 | 6 | 7 |
|---|---|---|---|---|---|---|---|---|
| Developed by/for | | ARYA | FEMA | NDMA | IIT-B | HPSDMA | BMTPC | MH-RVS (Proposed) |
| Source | | Arya, 2006 | FEMA, 2015 | NDMA, 2020 | Sinha & Goyal, 2004 | Kumar et al., 2016 | BMTPC, 2019 | Author |
| Typology | A1: Mud & Unburnt Brick | | | □ | □ | | □ | □ |
| | A2: Stone Wall | □ | | □ | □ | □ | □ | □ |
| | B: Burnt Brick | □ | □ | □ | □ | □ | □ | □ |
| | C1: Concrete Wall | □ | □ | □ | □ | □ | □ | □ |
| | C2: Wood Wall | | □ | | □ | | □ | □ |
| | X: Other Materials | | | □ | | | □ | □ |
| | Steel | □ | □ | | □ | | | □ |
| General Information | About Building and owner | □ | □ | □ | □ | □ | | □ |
| | Sketch/Photo and drawings | □ | □ | | □ | | | □ |
| | Occupancy (Day & Night) | □ | □ | | □ | □ | | □ |
| | Cost of Construction | | | | | □ | | |
| | Construction quality and Maintenance | | □ | □ | | | | □ |
| Disaster History | Seismic Zone | | □ | □ | □ | | □ | □ |
| | Disaster History and Damage status | | | | | □ | | □ |
| | Disaster cause | | | | | □ | | |
| | Retrofitting history | | | | | | | □ |
| Site Condition | Location of building | | | | □ | | | □ |

| Category | Parameter | | | | | | | |
|---|---|---|---|---|---|---|---|---|
| | Site Condition | | | ☐ | | ☐ | | ☐ |
| Building Geometry | Dimension of Building | | | | | ☐ | | |
| | Shape of Building, floors | ☐ | ☐ | ☐ | ☐ | ☐ | | ☐ |
| | Re-entrant corners | | | | | ☐ | | ☐ |
| Foundation | Type of Sub-Soil | ☐ | ☐ | ☐ | ☐ | ☐ | | ☐ |
| | Foundation detail | ☐ | | | | ☐ | | ☐ |
| | Depth of ground water table | ☐ | | ☐ | | ☐ | | ☐ |
| Walls | Walls details | ☐ | ☐ | ☐ | | ☐ | ☐ | ☐ |
| | Separation of walls at joint | | | ☐ | | | | ☐ |
| | Wall failure observed | | | ☐ | | ☐ | | ☐ |
| Earthquake Bands | Earthquake band details and status | | | ☐ | | ☐ | | ☐ |
| Cracks | Cracks details | | | ☐ | | ☐ | | ☐ |
| | grade of cracks | ☐ | | ☐ | | ☐ | | ☐ |
| Openings | Opening(s) details | | | ☐ | | ☐ | | ☐ |
| | Frames details near opening | | | | | | | ☐ |
| Roof and Floor | Type and material | | ☐ | ☐ | | ☐ | ☐ | ☐ |
| | Roof's attachment with walls | | | ☐ | | ☐ | | ☐ |
| | Failures observed | | | | | ☐ | | ☐ |
| Pounding effect | Height of building | | | ☐ | | ☐ | | ☐ |
| | distance from closest building | | | | | | | ☐ |
| | Quality of adjacent building | | ☐ | ☐ | | ☐ | | ☐ |
| Heavy weight on top | Type and positioning of Heavy weights | | | | | ☐ | | ☐ |
| | Intact status with structure | | | | | | | ☐ |
| Parapet | Parapet material | | | ☐ | | ☐ | | ☐ |
| | Parapet intact with structure | | | ☐ | | | | ☐ |
| Overhang | Type of overhangs | ☐ | ☐ | ☐ | ☐ | ☐ | | ☐ |
| | length and intact status | | | ☐ | | | | ☐ |
| Staircase | Staircase details | ☐ | | ☐ | | ☐ | | ☐ |
| | Lift status | | | | | | | ☐ |
| Column and Beam | Column Beam details | | | ☐ | | ☐ | | ☐ |
| | Beam with infill wall | | ☐ | | | | | ☐ |
| | Connection and continuity | ☐ | | ☐ | | | | ☐ |
| Basement | No. of basement | | | | | ☐ | | ☐ |
| | Column and retaining Wall | | | | | | | ☐ |
| Soft Storey | Soft Storey's details | | ☐ | ☐ | | ☐ | | ☐ |
| High Wind | Potential threat from wind | | | | | | | ☐ |
| Landslide | Position of potential landslide | ☐ | ☐ | ☐ | | | | ☐ |
| | Stabilized slope status | | ☐ | ☐ | | | | ☐ |
| | Barriers to rockfall | | | ☐ | | | | ☐ |
| Industrial | Potential threat from Industrial Hazard | | | | | | | ☐ |
| Fire | Fire Safety Status | | | | | ☐ | | ☐ |

| | | | | | | | |
|---|---|---|---|---|---|---|---|
| | Location of potential fire threats | | | | | | | ☐ |
| Climate Change | Understanding & Concern | | | | | | ☐ |
| Non-Structural Elements | Cantilever availability (Chimneys, Balconies, Parapet, Sunshades, claddings) | ☐ | ☐ | ☐ | ☐ | ☐ | | ☐ |
| | Other Non-Structural elements | ☐ | ☐ | ☐ | ☐ | ☐ | | ☐ |
| | No. of unattached Non-structural elements | | | | | | | ☐ |
| | | | | | | ☐: Concern (major/minor) | | |


# 4   IHR Specific MHRA Survey Form Preparation

*4.1   Survey Form Preparation*

The proposed survey form is a modification of the Rapid Visual Screening (RVS) survey questionnaire, i.e., a form used for structural and non-structural components of a building that performs during an Earthquake. In the original RVS questionnaire no other hazards are considered. A building's location on a vulnerable site, its structural condition, and performance can lead to disastrous situations. The other hill-specific hazards are also incorporated into the proposed form to identify the risk components from multi-hazards. Whilst the Himalayan region is prone to earthquakes as per India's Seismic Zonation Map (Fig. 3), the proposed survey form also covers other hazards like landslide, flood, industrial explosion/emissions, fire vulnerability, hydro-climatic factors, etc., which will be addressed one by one in this paper.

*4.2   Preliminary Survey*

Before conducting the Pilot survey, a preliminary survey has been conducted to test the proposed form, research methodology, and identifying gaps in the existing survey form.

This small assessment also evaluated the RVS form with minor enhancements to evaluate its performance and confirm gaps, and to see if it can meet the requirement for risk assessment at other areas with similar geographical characteristics and conditions as experienced in the Indian Himalayan Region.

The Preliminary survey was conducted at 5 Gram Panchayats of Chinyalisaur sub-district in Uttarkashi, Uttarakhand, namely Chinyalisaur, Dhanpur, Dharasu, Hidhara, and Bagi, in October and November 2019, using Draft MHRA Survey form. Some of the pictures of the visit are provided in ig5.

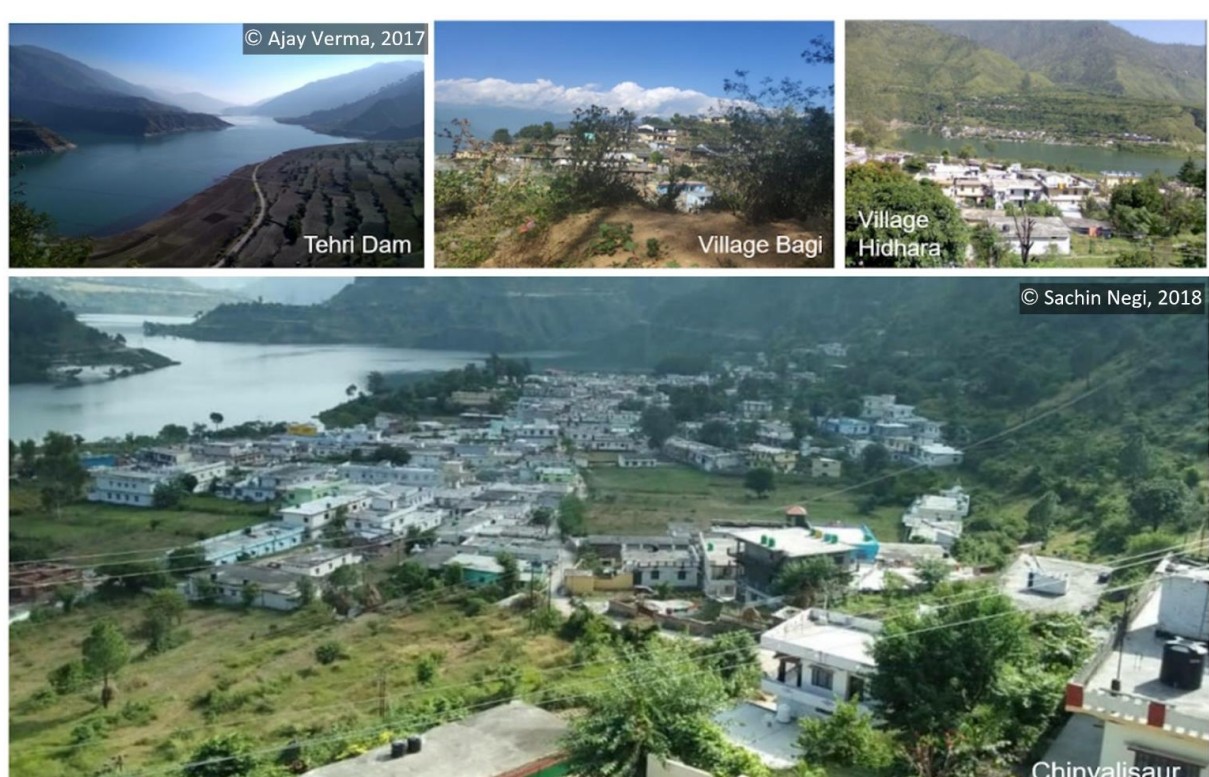

Figure 5: View of Site selected for Pilot Survey

The preliminary survey was conducted to determine (1) Whether the questions are clearly framed? (2) Does it cover all the requirements as per hill communities? (3) Is the wording of the questions correct enough to lead to the desired outcomes? (4) Are the questions as well options for answers suggested hill specific or not? (5) Are the questions positioned in the most satisfactory order? (6) Do surveyors and respondents of all classes understand the questions? (7) Are the questions and their options self-explanatory or not? (8) Do the sections in the survey form cover risk assessment related questions for all identified hazards or not? (9) Are the questions as per construction practices and construction materials available on hills or not? (10) Is there any need to add some questions or specific, or do some need to be eliminated so as to improve the flow of the survey session. (11) Do the surveyor and respondent understand the importance of this survey or the objective behind this survey and responded in that way?

### 4.2.1  Observations during Preliminary survey

Feedback from the Preliminary study proved very helpful in determining the key gaps and shortcomings of the form design and in informing improvements to the proposed form design. Specifically (1) The preliminary study showed that a surveyor's observations of a project site, his or her understanding of each question, and his/her strategy for convincing the residents to provide accurate data played a significant role in risk assessment. (2) In some questions, the use of technical terms or difficult words, or questions designed to gather too much data at

once, discourage respondent interest in responding further and make the Surveyor
uncomfortable to proceed. (3) The questionnaire may not be self-explanatory and requires
someone with civil engineering training to fill it out. (4) Building geometry, construction
practices, construction materials, and development trends play an essential role during any
hazard, thus existing building related questions and options must be incorporated. (5) Survey
questions are developed primarily from observations made by survey and engineers as
opposed to responses from residents. (6) If the Surveyor is not familiar with the terminologies
and aims behind filling out the questionnaire, it leads to no response or respondents
sometimes loose interest to answer further. (7) An unclear survey vision, study purpose, and
inadequate training of the Surveyor will make it difficult to explain the importance of data
collection to the respondent, leading to unclear questions and less accurate responses. (8)
Surveyors should be trained enough to pick out the correct option from respondents' lengthy
responses. (9) Need of pictorial representation of answers/options for better understanding of
the Surveyor. (10) Different answers are obtained when questions are arranged
inappropriately or answers are arranged incorrectly. (11) Observing the interaction between
multiple hazard types in the same area is a challenging aspect of natural hazards risk
assessment.
*4.3  Proposed MHRA Form*
After the Preliminary survey conducted at the Chinyalisaur sub-district, significant points were
identified/observed that has been incorporated in the Proposed survey form of Multi-Hazard
at hill locations with all the simple content and graphical inputs for better understanding.
Hence, the modifications from a Multi-hazard risk point of view and surveyors' point of view
can be seen in the proposed form (Table 5 and 6).
These amendments and the full survey form are presented below.
*Table 5a: Proposed MHRA Survey form (Part A)*

| Rapid Visual Screening (RVS) form | | |
|---|---|---|
| **SURVEYOR** | | |
| 1 | Name of the Surveyor | |
| 2 | Mobile no. of Surveyor | |
| 3 | Inspection Data | |
| 4 | Inspection Time | |


| GENERAL INFORMATION | | | | | |
|---|---|---|---|---|---|
| 5 | Name of Building/Owner | | | | |
| 6 | Address | | | | |
| 7 | Town/City, District and State | | | | |
| 8 | Coordinatnates | | | | |
| 9 | Total No. of Building Blocks present inpremises | | | | |
| 10 | Name of Block to be survey | | | | |
| 11 | Draw Sketch of Site Plan | | | | |

| 12 | Function of Block | Residential (Individual House) | | Residential (Appartments) | | Residential (Other) |
|---|---|---|---|---|---|---|
| | | Educational (School) | Educational (College) | Educational (Institute/ University) | | |
| | | Lifeline (Hospital) | Lifeline (Police Station) | Lifeline (Fire Station) | Lifeline (Power Station) | Lifeline (Water/ Sewage Plant) |
| | | Commercial (Hotel) | Commencial (Shopping) | Commercial (Recreational) | | Commercial (Other) |
| | | Office (Govt.) | | Office (Private) | | |
| | | Mixed Use (Residential and Commercial) | | Mixed Use (Residential and Induustrial) | | Mixed Use (Other) |
| | | Industrial (Agriculture) | | Industrial (Live Stick) | | Industrial (Other) |
| 13 | Occupancy in day time | 0 to 10 | 11 to 50 | 51 to 100 | 101 to 1000 | more than 1000 |
| 14 | Occupancy in night time | 0 to 10 | 10 to 20 | 51 to 100 | 101 to 1000 | more than 1000 |
| 15 | Name of Owner | | | | | |
| 16 | Name of Contact Person | | | | | |
| 17 | Contact No. of Contact Person | | | | | |
| 18 | Year of Construction: | | | | | |
| 19 | Structural or Construction drawings available? | Yes | | No | | |
| 20 | Total built up area (sq.m) | | | | | |
| 21 | No. of Floors | Low Rise (1 to 3) | Mid Rise (4 to 7) | | High Rise (7 and above) | |
| 22 | What is the overall Construction quality | Excellent | Good | Average | Poor | Very Poor |
| 23 | What is the overall Maintainance Status | Excellent | Good | Average | Poor | Very Poor |




| | | DISASTER HISTORY | | | | |
|---|---|---|---|---|---|---|
| 24 | Seismic Zone | Zone V | Zone IV | Zone III | Zone II | Don't know |
| 25 | Did this area faced any Major disaster?: | Yes | | No | | |
| 26 | If Yes in Q.25, Which Disaster?: | Earthquake | Flood | Landslide | Wind | Industrial |
| | | | | | | |
| | | Fire | Other | If Other, Specify | | |
| | | | | | | |
| 27 | If Yes in Q.25, in which date/year | | | | | |
| 28 | If Yes in Q.25,What is the major damage status | No effect | Minimum Effect | Medium Effect | Maximum Effect | |
| | | | | | | |
| 29 | Is the building Retrofitted/ Renovated ever? | Yes | | No | | |
| 30 | If Yes in Q.29, Year of last renovated? | | | | | |

**501**

| | | SITE CONDITION | | | |
|---|---|---|---|---|---|
| 31 | Location of Building: | Isolated | Internal Corner | | End |
| | |  House |  H | |  H |
| | | | | | |
| 32 | Slope of Ground: | Flat Terrain | Gentle Slope | Steep Slope | Terraced land |
| | |  |  |  |  |
| | | | | | |
| 33 | Cut & Fill Material: | RCC | Hybrid | | Other |
| 34 | Is there Visible cracks on the ground | Yes, Many | | Yes, few | No |
| 35 | Is there any open space in the property? | Yes, more than 1500 sq.ft | | Yes, less than 1500 sq.ft | No |
| 36 | What is the total area of Open spaces in the campus (in sq.ft) : | | | | |

**502**
**503**

Note: RCC: Reinforced Cement Concrete; H: House position

| | | BUILDING GEOMETRY | | | | |
|---|---|---|---|---|---|---|
| 37 | Shape of Building Block in Plan: | Square | Rectangle (L<=3B) | Narrow Rectangle (L>3B) | Rectangle with courtyard | L-Shaped |
| | |  |  |  |  |  |
| | | | | | | |
| | | T-Shaped | U-Shaped | E-Shaped with Central courtyard | H-Shaped | Other |
| | |  |  |  |  | |
| | | | | | | |

**504**

| 38 | Shape of building Block in Elevation: No. of Reentrants corner in Plan | Not stepped | Stepped near centre | Stepped near the end | Heavy upper floor | |
|---|---|---|---|---|---|---|
| | | 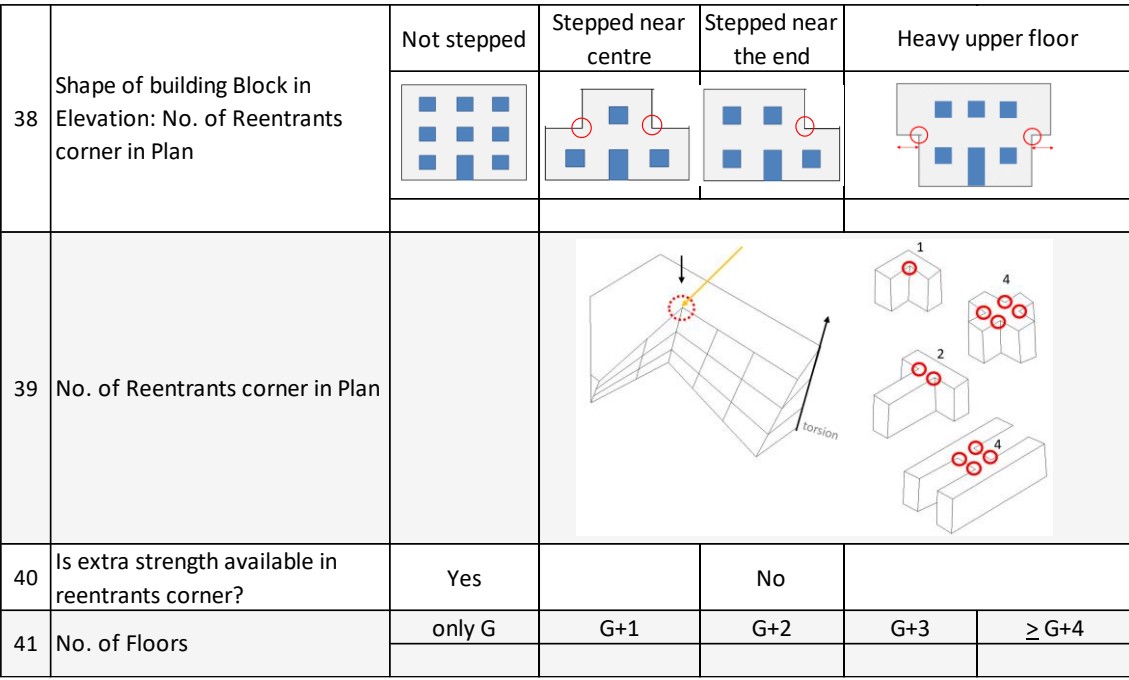 | | | | |
| | | | | | | |

| 39 | No. of Reentrants corner in Plan | | | |
|---|---|---|---|---|
| | | | | |

| 40 | Is extra strength available in reentrants corner? | Yes | | No | | |
|---|---|---|---|---|---|---|

| 41 | No. of Floors | only G | G+1 | G+2 | G+3 | ≥ G+4 |
|---|---|---|---|---|---|---|
| | | | | | | |

Note: G: Ground floor

| FOUNDATION | | | | | |
|---|---|---|---|---|---|
| 42 | Type of Sub Soil: | Rock | Gravel or Sand | Soft or Medium | Other |
| | | | | | |
| | | | | | |
| 43 | Type of Foundation: | Strip | | Raft | Isolated |
| | | | | | |
| | | Pile | | Combined | Other |
| | | | | | |
| | | | | | |


| 44 | Basic Construction material of Foundation: | Adope | Stone | Brick | RCC | Other |
|---|---|---|---|---|---|---|
| | | |  |  |  | |
| | | | | | | |

| 45 | Mortar Material in Foundation: | Dry Masonry | Mud | Lime | Cement | Other |
|---|---|---|---|---|---|---|
| | | | | | | |

| 46 | Plinth beam available? | Yes | No |  |
|---|---|---|---|---|
| | | | | PLINTH BAND |

| 47 | Sinking in Foundation? | Yes | | Partial | | No | |
|---|---|---|---|---|---|---|---|
| | | | | | | | |

| 48 | If Yes or Partial in Q.47, What is the Reason for Sinking? | Cause of nearest water resources | | Without any water resources | | Other (specify) | |
|---|---|---|---|---|---|---|---|
| | | | | | | | |

| 49 | Depth of ground water table | | | | | Don't know |
|---|---|---|---|---|---|---|

**WALL**

| 50 | Type of Wall: | Brick | Stone | Confined | RCC | Other |
|---|---|---|---|---|---|---|
| | |  |  | Only Column available & No Beams | Column & Beam, both available | |
| | | | | | | |

| 51 | Is through-stone used in Stone Wall? | Yes | Partial | No |  |
|---|---|---|---|---|---|
| | | | | | Through Stone |

| 52 | What is the Wall material? | Adobe or Mud Wall | River Boulder wall | Quarry Stone wall | Dressed wall | fired brick wall |
|---|---|---|---|---|---|---|
| | |  |  |  |  |  |
| | | | | | | |
| | | hollow concrete block wall | | | Other | |
| | |  | | | | |
| | | | | | | |



| 53 | Type of mortar | Dry masonry | Mud | Lime | Cement | Other |
|---|---|---|---|---|---|---|
| | | | | | | |
| 54 | Thickness of interior Wall (in mm): | < 115 mm | 115 mm (4.5") | 230 mm (9") | 230 to 450 mm | > 450 mm |
| | | | | | | |
| | Length of longest interior wall (in meter) | | | | | |
| | Max. Height of the wall (in meters) | | | | | |
| 55 | Thickness of exterior Wall (in mm): | < 115 mm | 115 mm | 230 mm | 230 to 450 mm | > 450 mm |
| | | | | | | |
| | Length of longest exterior wall (in meter) | | | | | |
| 56 | Thickness of Mortar (in mm): | | | | | |
| 57 | How many Separation of walls at T and L junction? | | | | | |


| 58 | Wall Failure type observed: | Bulging of wall | delaminating of wall | tilting of walls | dampness in wall | No failure |
|---|---|---|---|---|---|---|
| | | | | | | |
| | No. of walls with these failures | | | | | |

Note: RCC: Reinforced Cement Concrete

| EARTHQUAKE BANDS | | | | | |
|---|---|---|---|---|---|
| 59 | Which of the Earthquake bands available? | Plinth Band | Sill Band | Lintel Band | Roof Band | |
| | | | | | | |
| | | Gable Band | Door Band | Window Band | Corner Band | No Band |
| | | | | | | |
| 60 | If Bands available in Q.59, What is the Material of Band: | Wood | Reinforced brick | Reinforced concrete | Other (Specify) | |
| 61 | If Bands available in Q.59, Thickness of Band (in mm): | | | | | |
| 62 | If bands available in Q59, Are the bands continuous? | Yes | Partial | No | | Don't know |


| | | CRACKS | | | |
|---|---|---|---|---|---|
| | Type of Cracks: | Structural cracks | Superficial cracks | | N/A |
| 63 | Note: Superfial cracks are seen in one side of wall, on the other hand structural cracks can be seen on both side of the wall |  |  | | |

| | | Diagonal cracks | Vertical cracks | Horizontal Cracks | Remark | |
|---|---|---|---|---|---|---|
| 64 | Type of Structural cracks: |  |  |  | | |
| | Specify, No. of Cracks in each case | | | | | |
| | Specify, Length of cracks in each case (in cm) | | | | | |
| | Grade of Cracks | Grade 5 | Grade 4 | Grade 3 | Grade 2 | Grade 1 |
| 65 | Are there any cracks on | Column | Beam | Near Openings | Near corner | No cracks |

| | | OPENING | | | |
|---|---|---|---|---|---|
| 66 | Is there any opening(s) larger than 50% of the length of the wall | Yes, all | | Yes, few | No | |
| 67 | Are there any opening close to wall junction or corner or to floor/roof | Yes, all | | Yes, few | No | |
| 68 | Is frames available around the door?: | Yes | | Partial | No | |
| 69 | If Yes/Partial in Q.68, What is the material of Frame used: | Wooden | MS/SS | other (Specify) | | |
| 70 | Is frames available around the window | Yes | | Partial | No | |
| 71 | If Yes/Partial in Q.70, What is the material of Frame used: | Wooden | MS/SS | other (Specify) | | |
| 72 | Is Grills available around the window?: | Yes | | Partial | No | |

Note: MS: Mild Steel, SS: Stainless Steel

| ROOF AND FLOOR | | | | | | |
|---|---|---|---|---|---|---|
| 73 | Type of Roof: | Flat Roof | One side slope | two side slope | four side slope | Other (specify) |
| | |  |  |  |  | |
| | | | | | | |
| 74 | Material of Roof: | RCC | | Reinforced brick slab | Tile or slate | CGI Sheets |
| | |  | |  |  |  |
| | | | | | | |
| | | Jack arch roof | | Wooden | Other (Specify) | |
| | |  | |  | | |
| | | | | | | |
| 75 | Are the roof anchored into the wall | Yes | | Partial | No | |
| | | | | | | |
| 76 | Type of Roof failures observed | Sagging | Cracks | Dampness | Other | No failure |
| 77 | Type of Flooring | Mud | Stone | Concrete | Wood.bamboo | Mosaic floor tile |

Note: RCC: Reinforced Cement Concrete; CGI: Corrugated Galvanized Iron

| POUNDING EFFECT DETAILS | | | | |
|---|---|---|---|---|
| 78 | Height of Structure /Block (in meters) | | | |
| 79 | Is there any adjacent building, which is very close (no gaps) to this building | Yes | with very little gap | No |
| 80 | Distance from nearest buildings (in meters) | | | |
| 81 | Quality of adjacent building | Very Good | Good | Moderate | Poor | Very Poor |

| HEAVY WEIGHT ON TOP | | | | | |
|---|---|---|---|---|---|
| 82 | Type of Heavy weight present on the top of the building? | water tank (Concrete) | Water tank (Plastic) | Car Parking on the top of the building | Big hoarding |
| | | | | | |
| | | Heavy generator/ machine | Communication tower | Roof top Garden | Other | None |
| | | | | | |
| 83 | If Yes in Q.82, What is the Position of Heavy weight? | Centric | Eccentric | Distributed | Corners | Remark |
| | |  |  |  |  | |
| | | | | | | |
| 84 | Are the heavy weight intact properly with structure? | Yes | | Partial | No | |
| | | | | | | |

| PARAPET WALL | | | |
|---|---|---|---|
| 85 | Is Parapet wall present at roof | Yes | Partial | No |
| 86 | If Yes or Partial in Q.85, What is the Material of Parapet Wall? | Lightweight (Wooden, MS/SS) | Heavy weight (RCC, Brick) | Remark |
| 87 | Intact with structure | Yes | Partial | No |

Note: MS: Mild Steel, SS: Stainless Steel, RCC: Reinforced Cement Concrete

| OVERHANGS | | | |
|---|---|---|---|
| 88 | Overhangs present | Yes | No |
| 89 | Length of overhangs (meters) | | |
| 90 | Overhangs with structural | Yes | No |
| 91 | Overhangs with Brackets /beam | Yes | No |


| STAIRCASE | | | | | |
|---|---|---|---|---|---|
| 92 | Staircase present | Yes | | No | |
| 93 | Staircase placed at symmetrical location in plan of the bulding | Symmetrical | | Un-symmetrical | |
| 94 | If Yes in Q.92, What is the Material of Staircase? | RCC | Brick | Wooden | MS/SS | Other |
| 95 | If Yes in Q.68, Is Staircase intact with building structure? | Yes | | No | |
| 96 | Lift Status? | Intact | Not Intact | Not Available | |

Note: MS: Mild Steel, SS: Stainless Steel, RCC: Reinforced Cement Concrete

| COLUMN | | | | | |
|---|---|---|---|---|---|
| 97 | Column available? | Yes | | No | |
| 98 | If yes in Q.97, What is the type of Column? | Short Column | | Long Column | |
| 99 | Material of Column | Concrete | Masonry (Brick/ Stone) | Wood | Steel | Other |


| BEAM | | | | |
|---|---|---|---|---|
| 100 | Beam available? | Yes | | No |

| | | Yes | Partial | No |
|---|---|---|---|---|
| 101 | If Yes in Q.100., Beam with infill walls available? |  Infill Wall | |  Column, Beam, No Wall |

| | | Centric | Eccentric | Other |
|---|---|---|---|---|
| 102 | If Yes in Q.100., Beam – Column connections? |  Beam, Column, Centric Beam Column Joints |  Beam, Column, Eccentric Beam Column Joints | |

| | | Centric | Eccentric | Other |
|---|---|---|---|---|
| 103 | Beam -Beam Connection? | Centric | Eccentric | Other |

| | | Concrete | Masonry (Brick/ Stone) | Wood | Steel | Other |
|---|---|---|---|---|---|---|
| 104 | If Yes in Q.100., Material of Beam | Concrete | Masonry (Brick/ Stone) | Wood | Steel | Other |


| BASEMENT | | | |
|---|---|---|---|
| 105 | Is Basement Available? | Yes | No |
| 106 | If Yes in Q.105, No. of Basement | | |

| | | Short Column | Long Column |
|---|---|---|---|
| 107 | Effective height of column in basement? |  Height of Column, Crushing Failure, X : Y ≤ 1 : 12, X: Area of Column, Y: Height of Column |  Height of Column, Buckling Failure, X : Y ≤ 1 : 12, X: Area of Column, Y: Height of Column |

| | | Yes | No |
|---|---|---|---|
| 108 | Retaining wall available ? | Yes | No |

| | | RCC | Brick | Stone | Other |
|---|---|---|---|---|---|
| 109 | If Yes in Q.108, What is the Material of the retaining wall ? | RCC | Brick | Stone | Other |


Note: RCC: Reinforced Cement Concrete

| SOFT STOREY | | |
|---|---|---|
| A soft story building is a multi-story building in which one or more floors have windows, wide doors, large unobstructed commercial spaces, or other openings in places where a shear wall would normally be required for stability as a matter of earthquake engineering design. | 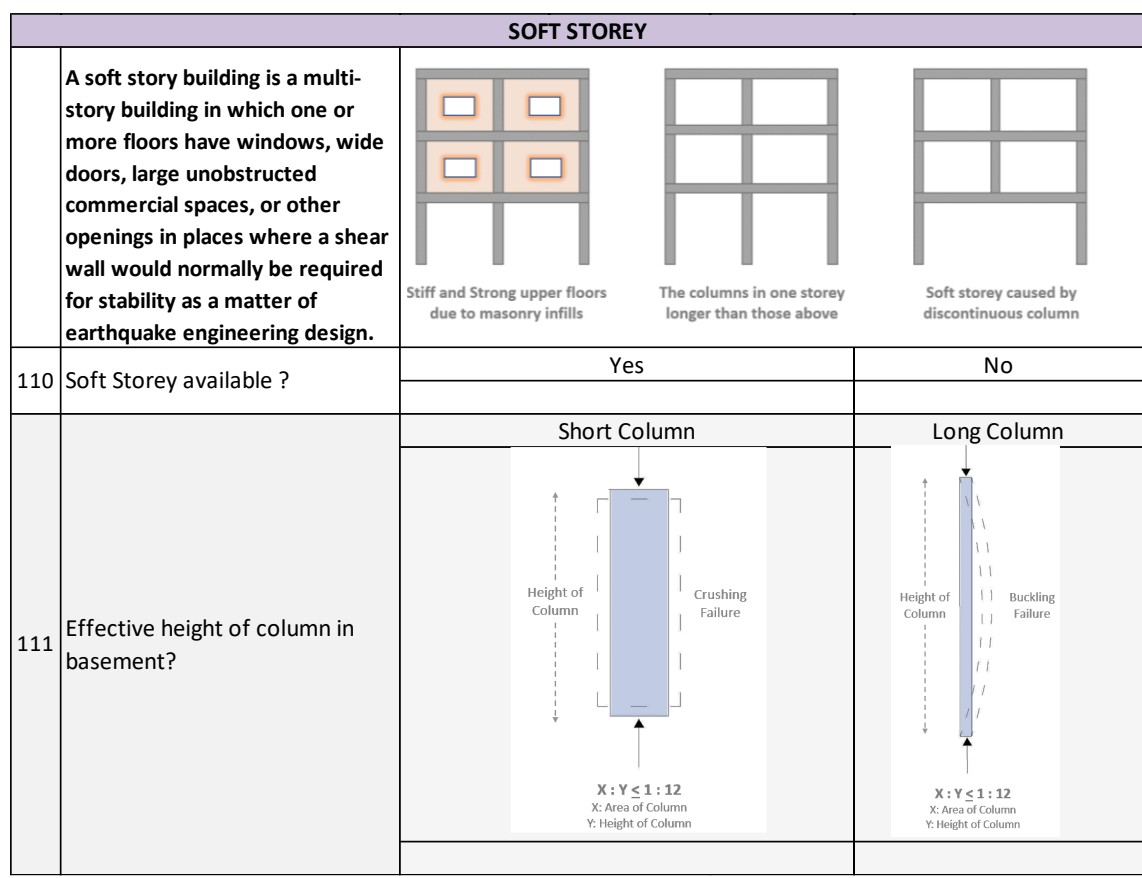 Stiff and Strong upper floors due to masonry infills / The columns in one storey longer than those above / Soft storey caused by discontinuous column | |

| | | Yes | No |
|---|---|---|---|
| 110 | Soft Storey available ? | | |

| | | Short Column | Long Column |
|---|---|---|---|
| 111 | Effective height of column in basement? | Height of Column / Crushing Failure / X : Y ≤ 1 : 12 / X: Area of Column / Y: Height of Column | Height of Column / Buckling Failure / X : Y ≤ 1 : 12 / X: Area of Column / Y: Height of Column |

| | | Yes | Partialy | No |
|---|---|---|---|---|
| 112 | Is shearwall available in Soft Storey? | | | |

| | | Yes | | No |
|---|---|---|---|---|
| 113 | Retaining wall available ? | | | |

| | | RCC | Brick | Stone | Other |
|---|---|---|---|---|---|
| 114 | If Yes in Q.113, What is the Material of the retaining wall ? | | | | |

*Table 6a: Proposed MHRA Survey form (Part B)*

| MULTI HAZARD SURVEY FORM | | | | | |
|---|---|---|---|---|---|
| **FLOOD** | | | | | |
| 1 | Is the site low lying or prone to water logging? | Yes | | No | |
| 2 | Is there any water body near the site? | Yes | | No | |
| 3 | What is the type of water body and whether it is prone to flooding? | Lake, flood prone | Lake, not flood prone | River, flood prone | River, not flood prone | N/A |
| 4 | What is the distance from the nearest water body? | 0 - 250 M | 250 - 500 M | 500 - 1000 M | 1 KM - 2 KM | 2 KM and above |
| 5 | What is the potential damage level due to the expected duration of flooding? | Very High | High | Medium | Low | Very Low |
| 6 | Is the plinth made up of non-erodible material? | Yes | | No | |
| 7 | What is the height of the plinth? (in meters) | | | | | |


| | HIGH WIND | | | | |
|---|---|---|---|---|---|
| 8 | What is the average wind speed in this location | Maximum Speed | | Minimum Speed | |
| 9 | Are there trees and/or towers too close to the building that may fall on it during high wind/cyclone? | can stop building from functioning | | threat can damage building but not hamper functioning | No threat |
| | | | | | |
| 10 | Do the door and windows have a good and accessible latch? | if neither doors or windows have accessible and good latches. | | If some of the doors and windows have accessible and good latches | If both doors and windows have accessible and good latches |
| | | | | | |
| 11 | Is there a covered walkway for building to building connection? | no covered walkway | | weak covered walkway | strong covered walkway |
| | | | | | |


| | LANDSLIDE | | | | |
|---|---|---|---|---|---|
| 12 | Is there any hills near to the building, which can cause damage due to landslide | Yes | | | No |
| | | | | | |
| 13 | If Yes in Q.12, what is the distance of the base off the Hill from building? | Less Than 30 M | 30 M - 100 M | 100 - 250 M | 250 - 500 M | More than 500 M |
| | | | | | |
| 14 | Is the slope near the building stabilized? | Yes | | | No |
| | | | | | |
| 15 | Are there any large rocks or potential falling hazards near the building? | Yes | | | No |
| | | | | | |
| 16 | Are there barriers to rockfall ? | Yes | | | No |
| | | | | | |


| | INDUSTRY | | | | |
|---|---|---|---|---|---|
| 17 | Is there any industry near to the building, which can cause damage due to industrial hazard, fire etc. | Yes | | | No |
| | | | | | |
| 18 | If Yes in Q.17, how many active industries are there? | Yes | | | No |
| | | | | | |
| 19 | What is the distance of nearest Industry from building? | 0 - 100 M | 100 - 250 M | 250 - 500 M | 500 - 1000 M | More than 1 km |
| | | | | | |
| 20 | What is the distance of nearest Petrol Pump from building? | 0 - 100 M | 100 - 250 M | 250 - 500 M | 500 - 1000 M | More than 1 km |
| | | | | | |


| FIRE | | | |
|---|---|---|---|
| 21 | Are the access roads from main street wide enough to allow one fire engine to reach, reverse and return to the main road? | **two or more such access roads**<br> | **one such access road**<br> | **No access road**<br> |

| 22 | Are there potential fire threats within 30 meters of the building such as petrol pump, electrical substation, combustible materials store, etc.? | Yes | | No |
|---|---|---|---|---|
| | | | | |

| 23 | Is there adequate open assembly area for people during any emergency? | enough space | inadequate open space (1-4 square feet per student) | negligible |
|---|---|---|---|---|
| | | | | |

| 24 | Is main meter box and switch box located in the staircase/ entrance lobby/ passage/ corridor? | Yes | No |
|---|---|---|---|
| | | | |

| 25 | Are the main meter box and switch box enclosed in a metallic box? | Yes | No |
|---|---|---|---|
| | | | |

| 26 | Is there more than 1 staircase which can be used as a fire escape staircase ideally at maximum distance from the other staircase? | Yes | No |
|---|---|---|---|
| | | | |

| 27 | In case of Public building or Life line building, Are there proper signages in the campus for Emergency Exit, Fire equipment etc.? | Yes<br> | No |
|---|---|---|---|
| | | | |

| 28 | Is the kitchen located at a safe distance from classrooms, staircase, passage corridor? | Yes, beyond 50 m | Yes, within 20-50 m | Yes, within 10-20 m | adjacent | Kitchen Not Available |
|---|---|---|---|---|---|---|
| | | | | | | |

| 29 | Is the ceiling material safe from fire? | Yes | No |
|---|---|---|---|
| | | | |

| 30 | What is the status of fire safety equipment in the building? | 100% - Fire extinguisher in each floor of each block | 75% - Fire extinguisher in 3/4th of all floors | 50% - Fire extinguisher in half of all floors | 25% - Fire extinguisher in 1/4th of all floors | 0% - No Equipment |
|---|---|---|---|---|---|---|
| | | | | | | |




| | | | | | | |
|---|---|---|---|---|---|---|
| 31 | Is the transformer too close to the compound wall or inside the building? | Yes | | | No | |
| 32 | Are there overhead cables running through or near premises/building? | Yes | | | No | |
| 33 | If there is a forest area near the building? | Yes | | | No | |
| 34 | What is the distance of the tree line from the building? | | | | | |
| 35 | Is there any combustible construction material present in the building? | Yes | | | No | |


| | CLIMATE CHANGE | | | | | |
|---|---|---|---|---|---|---|
| 36 | How much do you think climate change threatens your personal | Very Likely | Likely | Neutral | Unlikely | Very Unlikely |
| | | | | | | |
| 37 | Which issues are of more concern in your opinion? (On the scale of 10, more marks to most concerned) | Climate change/Global Warming | Poverty | Over-population | Un-employment | Crime |
| | | | | | | |
| | | Infectious Diseases | Economic Situation | Unplanned Infrastructure | Deforestation | Air pollution |
| | | | | | | |
| | | Water pollution | Tourism growth | Poor Waste Management | Extinction of species | Traffic |
| | | | | | | |
| 38 | In your opinion, What is the reason that the temperature on earth has been rising over the past decade? | Human Activities | Natural Causes | No Change | Don't know | Other |
| | | | | | | |
| 39 | How much do you think the following has contributed to global climate change? (on scale of 10, more marks to most contributer) | Deforestation | Overpopulation | Tourist growth | Landuse Landcover | Greenhouse gases |
| | | | | | | |
| | | Industrilization | Melting of Ice | Warming of water surface | Other | Don't know |
| | | | | | | |


| Non Structural Risk/ Falling Hazard | | | | | | |
|---|---|---|---|---|---|---|
| | | Element | Need Attentio | Number | Element | Need Attentio | Number |
| 40 | List of Nonstructural elements which are vulnerable to falling or not attached properly | Fan | | | Wooden Frame at Roof | | |
| | | Tubelight | | | Door | | |
| | | Electrical Wires | | | Window Frames | | |
| | | AC | | | Heavy Machinaries | | |
| | | Open Shelve (Glass) | | | Cylinder in Open space | | |
| | | Open Shelve (Iron) | | | Board | | |
| | | Wardrobe (Wooden) | | | Ventilator | | |
| | | Wardrobe (Iron) | | | Fire Extinguisher | | |
| | | HeavyTable | | | Cantilever Chimneys | | |
| | | Heavy Frames | | | Cantilever Balconies | | |
| | | Heavy Furnitures | | | Cantilever Sunshades | | |
| | | Heavy weight on top of almirah | | | Other | | |
| 41 | No. of Exits in the Room: | | | | | | |
| 42 | What is the status of Electrical Safety in the Room | GOOD | | | OK | POOR | |

## 4.4 Risk Score Computation

After all the parametric studies from various Indian Standard codes and Reports ((NDMA, 2020), (URDPFI, 2015); IS-13828 (1993); IS-4326 (1993); IS-1893-1 (2016); IS-13935, 2009; IS-15988 (2013)) on ideal building parameters and weak components of a building from the design, construction, site condition, surrounding condition, location and hazard points of view, risk scores were decided on an average basis on 24 components separately (refer section 4.5 of this paper) for better judgment and understanding. Risk scores were derived from the proposed survey form by appropriately weighing the data points against a risk number chart with higher weightage given to higher risk (Chouhan et al., 2022b). The data was then aggregated on a scale of ten (Table 7). For example, if a building answers all weighted MCQs with the highest risk option, it will be scored 10/10 and similarly for low risk and moderate risk. All questions in the questionnaire were not weighted; those with ambiguous risk consequences were left un-weighted to be studied objectively. The risk scores intend to give a relative idea of where the risk lies within a building and among buildings to enable prioritization during risk mitigation planning.

Table 7: Risk Score Computation, Source adapted from (Chouhan et al., 2022b)

| Risk Score | 0 to 2 | 2.1 to 4 | 4.1 to 6 | 6.1 to 8 | 8.1 to 10 |
|---|---|---|---|---|---|
| Risk Status | Very low | Low | Moderate | High | Very high |

| Building Status | Very Safe | Safe | Moderately Safe | Unsafe | Very Unsafe |
|---|---|---|---|---|---|
| **Recommendation** | Need Maintenance | Need Attention and Maintenance | Need Attention and SVA | Required DVA and Retrofitting | Required Retrofitting Urgently |
| | Under the supervision of experts | | | | |
| | *SVA: Simplified Vulnerability Assessment, DVA: Detailed Vulnerability Assessment* | | | | |


*4.5   Pilot Survey*
After finalization of the proposed MHRA Survey form, a Pilot survey was conducted at 10
schools of Uttarakhand state. The results of the building level survey and campus level survey
are shown below in section 4.5.1. and 4.5.2.
**4.5.1   Result of Rapid Visual Screening Survey**
As per IS Code 13935 (2009), the key goal of seismic reinforcement is to improve a weakened
building's seismic resilience as it is being repaired, making it stronger in the event of potential
earthquakes. The individual results of 17 components of RVS are elaborated, which highlights
the weaker part that needs attention in a building.
Table 8: *Result of RVS of 10 schools through Proposed form*

| SN | Risk Status | Very Low Risk | Low Risk | Moderate Risk | High Risk | Very High Risk | Total |
|---|---|---|---|---|---|---|---|
| 1 | Site Condition | 54 % | 13 % | 29 % | 2 % | 2 % | 100 % |
| | | 32 | 8 | 17 | 1 | 1 | 59 blocks |
| 2 | Building Geometry | 34 % | 27 % | 14 % | 20 % | 5 % | 100 % |
| | | 20 | 16 | 8 | 12 | 3 | 59 blocks |
| 3 | Foundation | 27 % | 22 % | 51 % | 0 % | 0 % | 100 % |
| | | 16 | 13 | 30 | 0 | 0 | 59 blocks |
| 4 | Wall | 36 % | 37 % | 27 % | 0 % | 0 % | 100 % |
| | | 21 | 22 | 16 | 0 | 0 | 59 blocks |
| 5 | Earthquake Bands | 0 % | 0 % | 7 % | 10 % | 83 % | 100 % |
| | | 0 | 0 | 4 | 6 | 49 | 59 blocks |
| 6 | Cracks | 2 % | 83 % | 0 % | 0 % | 15 % | 100 % |
| | | 1 | 49 | 0 | 0 | 9 | 59 blocks |
| 7 | Openings | 63 % | 17 % | 19 % | 1 % | 0 % | 100 % |
| | | 37 | 10 | 11 | 1 | 0 | 59 blocks |
| 8 | Roof | 7 % | 3 % | 10 % | 78 % | 2 % | 100 % |
| | | 4 | 2 | 6 | 46 | 1 | 59 blocks |
| 9 | Pounding Effect | 25 % | 0 % | 5 % | 39 % | 31 % | 100 % |
| | | 15 | 0 | 3 | 23 | 18 | 59 blocks |
| 10 | Heavy Weight on top | 95 % | 0 % | 2 % | 0 % | 3 % | 100 % |
| | | 56 | 0 | 1 | 0 | 2 | 59 blocks |
| 11 | Parapet | 93 % | 0 % | 7 % | 0 % | 0 % | 100 % |
| | | 45 | 0 | 4 | 0 | 0 | 59 blocks |

| 12 | Overhang | 53 % | 0 % | 15 % | 0 % | 32 % | 100 % |
| | | 31 | 0 | 9 | 0 | 19 | 59 blocks |
| 13 | Staircase | 80 % | 0 % | 3 % | 12 % | 5 % | 100 % |
| | | 47 | 0 | 2 | 7 | 3 | 59 blocks |
| 14 | Column | 51 % | 0 % | 12 % | 0 % | 37 % | 100 % |
| | | 30 | 0 | 7 | 0 | 22 | 59 blocks |
| 15 | Beam | 32 % | 2 % | 7 % | 7 % | 52 % | 100 % |
| | | 19 | 1 | 4 | 4 | 31 | 59 blocks |
| 16 | Basement | 100 % | 0 % | 0 % | 0 % | 0 % | 100 % |
| | | 59 | 0 | 0 | 0 | 0 | 59 blocks |
| 17 | Soft Storey | 100 % | 0 % | 0 % | 0 % | 0 % | 100 % |
| | | 59 | 0 | 0 | 0 | 0 | 59 blocks |


### 4.5.2  Result of Multi-Hazard Survey

The survey was conducted by considering the campus of the school as one unit. It primarily
focuses on the location of school premises under a vulnerable zone or not, if yes, to which
kind of hazard. It solves the question of how the school campus is prepared. The result of
multi-hazard survey is shown in the Table 9 below:
*Table 9: Result of Multi-Hazards of 10 schools through Proposed form*

| SN | Risk Status | Very Low Risk | Low Risk | Moderate Risk | High Risk | Very High Risk | Total |
|---|---|---|---|---|---|---|---|
| 1 | Flood Risk | 10% | 50% | 30% | 0% | 10% | 100% |
| | | 1 | 5 | 3 | 0 | 1 | 10 Schools |
| 2 | High Wind Risk | 70% | 20% | 10% | 0% | 0% | 100% |
| | | 7 | 2 | 1 | 0 | 0 | 10 Schools |
| 3 | Landslide Risk | 100% | 0% | 0% | 0% | 0% | 100% |
| | | 10 | 0 | 0 | 0 | 0 | 10 Schools |
| 4 | Industrial Risk | 100% | 0% | 0% | 0% | 0% | 100% |
| | | 10 | 0 | 0 | 0 | 0 | 10 Schools |
| 5 | Fire Risk | 0% | 20% | 60% | 20% | 0% | 100% |
| | | 0 | 2 | 6 | 2 | 0 | 10 Schools |
| 6 | Non-Structural Risk | 0% | 0% | 0% | 80% | 20% | 100% |
| | | 0 | 0 | 0 | 8 | 2 | 10 Schools |


The photos of the 10 schools where pilot survey was conducted is shown in the Fig. 6 below:

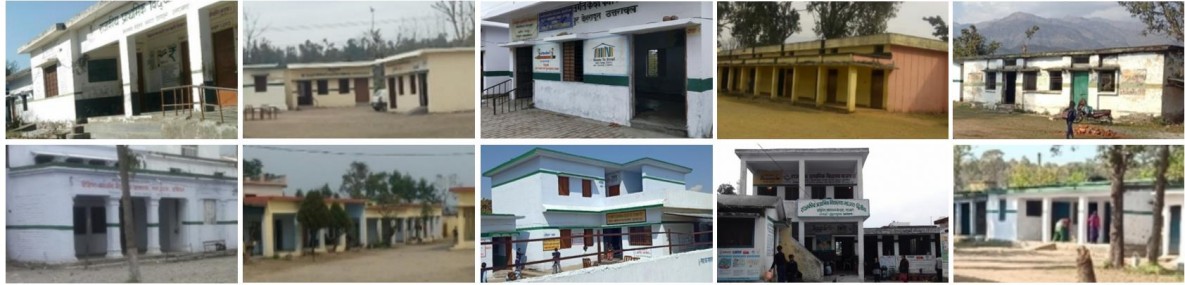

Figure 6: *Photo of the 10 schools*

## 5   Discussion:

*5.1   Pilot Survey*

The IHR requires effective and standardised Multi-Hazard Risk Assessment, and for that purpose a customized designed Survey Form has been designed to capture the unique characteristics of hill communities and assets. The proposed form performed reasonably well. Effectiveness & data collection is comfortable from both ends i.e., Respondents & Surveyor. The questions are properly framed in various sections, the language is simple and it is easy to interpret. The pictorial explanation makes it easy for surveyors to correct input data, as its explanation is self-explanatory. The objective behind the data collection is well clear to the Respondents and Surveyor.

*5.2   Key features of the proposed MHRA survey form*

The key features of the proposed form are it is specially designed for data collection in the Indian Himalayan region with risk of earthquake, flood, high wind, industrial hazard, non-structural risk, fire vulnerability  and climate change awareness. As the value addition, the proposed survey form consist of questions related to climate change also, as the promotion of self-mobilisation and action is enhanced by awareness; it increases enthusiasm and support. It is therefore crucial to raise awareness about climate change adaptation in order to manage the impacts of climate change, increase adaptive capacity, and reduce overall vulnerability.

The proposed survey form is very useful for any type of study related to Hazard Risk assessment in hills. Time taken to complete the questionnaire, i.e. the length of the questionnaire is good enough i.e. 10 minutes for the trained civil engineer and 17 minutes for the trained non-engineering background surveyor. With practice, the surveyor can reduce time. The language of the form is simple and specific, i.e. one answer on one dimension is required, it considers all possible contingencies when determining a response and it is designed in a way that it collects more & more accurate information in less time. Questionnaires permit the collection and analysis of quantitative data in a standardized manner, ensuring their internal consistency and coherence. The question sequence is clear and smooth moving. By sequencing questions properly, the chances of misinterpreting

individual questions are greatly reduced. The pictorial options make it comfortable for the
surveyor to fill the answer by looking at the building.
The survey form is divided into sections so that only one thought can be conveyed at a time.
It includes the advanced version of RVS that covers risk status for foundation, wall, roof,
openings, beam, column, site conditions, etc. of a building. It is covering all the points required
for building analysis in RVS. It covers questions related to all identified hazards that are directly
indirectly contributing to risk factors. It covers all the required questions as per hill condition,
situation, climate, geography, construction practices, construction materials, etc. The format,
including the font and layout, is good enough to read by the surveyor. Before going into the
field, the surveyor must require a reading of the full survey form carefully with all terminologies
clear. It includes non-structural risk survey questions. The safety of occupants in a building
following an incident can be at risk due to reduced capacity of structural components or
damage to non-structural components. This hill-specific MHRA questionnaire survey may act
as a risk sensitization tool.
*5.3    Result of Pilot Survey*
It can be seen that the detailed multi-hazard risk assessment will help the schools to identify
the potential threats presented in the building as well as premises and the steps to retrofit the
structure.
Due to the region's strong earthquake zonation, RVS and NSRA (Non-Structural Risk
Assessment) data suggest high structural and non-structural vulnerability in almost all the 10
schools (figure 7), which assumes greater significance. On the other hand, schools need to
improve their fire safety measurement and trainings. High wind and floods pose a prominent
moderate to high risk. Industry and landslides, on the other hand, pose no risk. The risk of fire
arises from a shortage of fire safety equipment and structural issues such as the absence of
an alternate staircase, the incorrect placement of fire-risk properties, etc. Fire disasters have
the potential to be catastrophic, but this should be a top priority as we advance. The wind is a
significant concern in this region because it is vulnerable to frequent windstorms. High-speed
winds pose a risk in the form of hazard trees/ towers, flying objects weakly latched
doors/windows.
Heavy furniture (tables, cabinets) and hanging electrical items/wire products face a
considerable risk of falling in the case of a tragedy in different rooms and labs. Falling hazards
can obstruct escape routes and injure people as they collide with them during minor seismic
shaking/earthquakes. When a disaster strikes, it's crucial for students and workers to have as
little disruption as possible during the critical reaction time. Mitigation measures primarily
involve simple fixes of non-structural elements with the structural element (wall and floor) and
are hence, for the most part, low-cost solutions.
Overall, the total risk is rated moderate on the risk scale considered by the authors after
structural and non-structural factors.

## 6   Conclusion

The Indian Himalayan region is facing disaster every year with significant loss of life and
property, as it is very prone to multi-hazards. Thousands of studies, research, and projects
are funded nationally and internationally to minimize the loss and prepare the community to
face the upcoming disaster.
A questionnaire is the backbone for any survey, which is the base for all types of research
work for better accuracy. This article describes why there is a need for a hill-specific survey
form that focuses on the multi-hazards in hills and hill's existing scenarios. It then described
the steps of how a Hill-specific Multi-Hazard Risk Assessment Survey form was developed,
validated through pilot survey, and tailored specifically for hill communities.
This article identifies gaps in the existing survey form used in India for risk assessment and
highlights the problem faced by the surveyors on ground while filling these survey forms. The
proposed form is a self-explanatory, pictorial, simple, easy to understand, covers hill specific
important components and it addresses several hazards such as earthquakes, floods, high
wind, landslides, industrial hazard, fire vulnerability and non-structural risk in the building.
The proposed survey form designed and applied under this study will help all the stakeholders
to collect better information from the field and made it easy for the surveyors to understand
even for non-technical person. This form will also identify the weak components of a building,
construction practices, their development trend, and vulnerability of the location, so that future
construction can be planned, considering the risk factors and vulnerable zones. Most of the
assessment criteria for multi-hazard risks are met by the proposed survey form. The more
accurate the data, the better will be its results.
The preliminary survey conducted at Chinyalisaur district of Uttarakhand validates the
questionnaire and survey form, and provided invaluable feedback now incorporated in to the
final survey form design. Through preliminary and pilot surveys it has been observed that the
proposed form is designed in a way that it can collect more accurate information in less time.
Questionnaires permit the collection and analysis of quantitative data in a standardized
manner, ensuring their internal consistency and coherence. The language and sequence of
questions is designed for clear and easy communication. Pictorial explanations of questions,
the unique feature, provides easy flow of information between the respondents and surveyors.
Thus, this hill-specific MHRA questionnaire survey may act as a risk sensitization tool.
The survey form is divided into various sections that covers firstly building specific questions
as buildings play crucial roles during any hazard, and secondly location specific questions that
cover the vulnerability of buildings towards other hazards. The result of the pilot survey
highlights the risk status for various components of a building which will help further in utilizing
the retrofitting and renovation budget in fruitful and planned way. On the other hand, the result
of the pilot survey also shows location wise vulnerability i.e., vulnerability of the building
towards other hazards that can help further in decision making related to disaster reduction,
preparedness and planning strategies at that location for that particular identified hazard. It
will also help to understand the development trend in that particular location and take action
for future development strategies.
The suggested form is a proposed version of Rapid Visual Screening (RVS), which can assess
the risk of any structure and includes all structural and non-structural components that respond
during a seismic event. It also includes information about the building's sensitivity to possible
danger zones such as landslides, floods, wind, and industrial hazards.
This study has the scope of application in other Asian countries with Himalayas like Nepal,
Bhutan, China and Pakistan. Its international application will enhance the survey form and
scope for future research. The proposed survey form will not only act as self-sensitization for
the building owners at micro level but will also have good scope at regional level i.e., macro
level, when results of all the buildings will be on single screen. The data collected using this
form can be used in any study related to Multi-Hazard Risk Assessment. It can be used by
civil engineers as well as non-civil engineering background people. People can self-assess
their building. To do this effectively, it is crucial to reinforce the networks of science,
technology, and decision-makers and create a sustainable technological outcome for disaster
risk reduction.
**Acknowledgment**
This research was supported by National Mission for Himalayan Studies (NMHS), Project
Grant No. NMH_1334_DMC and CoPREPARE, Project Grant no. IGP2020-24/COPREPARE
– funded by UGC. We are indebted to the local residents who actively participated in the
household survey. The authors are grateful to Mr. Tom Burkitt, from DHI, for supporting
editorial and proofreading.
**Data availability Statement**
This article is part of doctoral research and the data collection has been done by the first
author physically on-site. The data is available from the authors on the request basis.
**Disclosure statement**
No potential conflict of interest was reported by the authors.

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
