# Peer review of "Design and Application of a Multi-Hazard Risk Rapid Assessment Questionnaire for Hill Communities in the Indian Himalayan Region"

_Natural Hazards and Earth System Sciences, 2022_

## Referee Comment (RC2)

[referee-annotated manuscript omitted]

---

## Author Comment (AC2)

**Response to Referee#2**

**Comment (R2)-1:**

[Figure]

**Response:** We appreciate the reviewer's insightful suggestion. Taking this comment into consideration, we have modified the Introduction part in section 1 of the manuscript. However, Literature on RVS has already been mentioned in section 3.3.2. of the manuscript.

**Comment (R2)-2: Please ensure high quality figure 1 to 5**

**Response:** We thank the reviewer's for highlighting this point and accordingly, we have replaced all the figures (identified by referee-Figure 1 to 5) with high quality pixels. We will send the figures in separate files.

**Comment (R2)-3:**

[Figure]

**Response:** We appreciate the reviewer's insightful suggestion; accordingly, we have updated the Table 1.

**Comment (R2)-4:**

[Figure]

**Response:** We have refined the need of the study in section 2.3. However, advantage of the proposed procedure is already mention in section 5.2 of the manuscript.

**Comment (R2)-5:**

[Figure]

**Response:** We have revised the design methodology in Section 3.1 and detailed the overall methodology adopted in section 3.2 and figure 2 of the manuscript for better clarity.

**Comment (R2)-6:**

[Figure]

**Response:** We greatly appreciate the reviewer's thoughtful suggestion. Taking this comment into consideration, we have revised the methodology in section 3.2 and figure 2. We have also detailed it for better understanding and clarity on the overall methodology adoption.

**Comment (R2)-7:**

[Figure]

**Response:** We have revised the section 3.3.3 and incorporated it in the methodology figure 2.

**Comment (R2)-8:**

[Figure]

**Response:** We appreciate the reviewer's thoughtful suggestion. However, considering the structure of the manuscript, after a thorough discussion, we are continuing the flow of the structure as before, i.e. to combine all information related to literature reviews in section 3.3 of the manuscript, including information about RVS.

[Figure]

**Response:** We greatly appreciate the reviewer's thoughtful suggestion (which will definitely enhance our work) and we agree with it. Taking this comment into consideration, we have added Pilot Survey of 10 schools and its results in section 4.5 and discussion about its result in section 5.3 of the manuscript.

**Comment (R2)-10:**

[Figure]

**Response:** Taking this comment into consideration, we have revised the conclusion part.

We would like to thank the referees once again for taking the time to examine our manuscript. Our manuscript quality has been enhanced by your comments and suggestions.

---

## Author Response (AR1)

**Response to Referee#1**

**Note: Comment of Referee 1 is in Blue and Referee 2 is in Green colour**

We would like to thank the reviewers for their positive and insightful comments on the manuscript. Our responses to the comments are given below:

**Comment (R1)-1: Topic selected for study is appropriate, however, the treatment is not up to the mark.**

**Response:** We appreciate the reviewer for pointing this out. Taking this comment into consideration, we have revised the title as follow: "Design and Application of a Multi Hazard Risk Assessment Survey Questionnaire for the Indian Himalayan Region". Replaced 'testing' with 'application'.

**Comment (R1)-2: References cited are not correct and some references are missing.**

**Response:** Taking this important comment into consideration, we have corrected and included the missing references.

**Comment (R1)-3: Paper claims about multi hazard risk assessment, however, there is no explanation given on how various hazards and risks are integrated.**

**Response:** Taking this comment into consideration, we have added Results of Pilot Survey in section 4.5. for better clarity and improved the discussion on multi-hazard risk assessment in Section 5.3.

**Comment (R1)-4: Table 2 show the comparison of survey forms. Some of the hazards mentioned are not relevant to the methods listed, e.g.,**

**1)NDMA forms is only meant to earthquake risk, it has no mention of floods,**

**Response:** We appreciate the reviewer's insightful observation and we agree that NDMA forms have major concern towards earthquake risk, but NDMA forms also shows concern towards flood. In (NDMA, 2020) form under Soil & foundation conditions, it shows concern towards building built on river terrace, ground with high water table, liquefiable soil etc. i.e. multi-hazards.

**2) There is no mention of high winds in BMTPC form. It is suggested to mention only the objectives for which the individual forms have been generated.**

**Response:** We appreciate the reviewer's insightful suggestion. I would like to highlight that BMTPC (Refer Table 5- Damage Risk to Housing under Various Hazard Intensities of BMTPC, 2019) shows vulnerability of houses towards earthquakes, wind/cyclones, floods etc. Thus, this form includes concern for other hazards.

**Comment (R1)-5: Also, manuscript is largely in the report format i.e., with bullets and objective mentioned in the form of flow chart. It is suggested to follow research paper.**

**Response:** We have revised it in section 3.1, 3.3.2.1, 4.2 and 4.2.1 of the manuscript.

**Comment (R1)-6: References: Some of the links provided as references are not either not available or there no paper by that reference**

**E.g. 1)Pradesh, H., Pradeep, R. and Anoop, K. (2016) 'Rapid visual screening of different housing typologies', Natural  672  Hazards. Springer Netherlands. doi: 10.1007/s11069-016-2668-3.**

**Eg.2) Full author list is needed in the paper "Aksha, S. K. et al.(2020) 'A geospatial analysis of multi-hazard risk in', Geomatics, Natural Hazards and Risk. 604  Taylor & Francis, 11(1), pp. 88–111. doi: 10.1080/19475705.2019.1710580."**

**Response:** We have updated this section as per referencing format of the Journal. Some of the modifications are as follow:

Eg.1: Kumar, S. A., Rajaram, C., Mishra, S., Kumar, R. P., and Karnath, A.: Rapid visual screening of different housing typologies in Himachal Pradesh, India, Nat Hazards, 85(3), 1851-1875, doi: 10.1007/s11069-016-2668-3, (2016).

Eg.2: Sanam, K. A., Lynn, M. R., Luke, j., and Laurence, W. C. Jr.: A geospatial analysis of multi-hazard risk in Dharan, Nepal, Geomatics, Nat. Hazards Risk., 11(1), 88-111, https://doi.org/10.1080/19475705.2019.1710580, 2020.

**IS-1893 has been revised in 2016. Subsequently there were two amendments. However, authors still use 2002 version.**

**Response:** We would like to thank the reviewer for this positive evaluation. Taking this comment into consideration, we have added the IS Code 2016 provisions in section 4.4 of the manuscript as suggested.

**Comment (R1)-7: Authors have prepared a comprehensive multi-hazard form however; they have not indicated how the multi-hazard is computed.**

**Response:** We appreciate the reviewer for highlighting this point and we agree that step wise detail of multi-hazard risk computation is not part of the manuscript, as scope of Risk Calculation study by itself is huge and we have plan to detail it in separate article. Taking this comment into consideration, we have updated basic Multi-Hazard Risk Computation in section 4.4 and added Results of Pilot Survey in section 4.5. This will improve clarity about risk computation using this proposed Survey form. The aim behind this manuscript is to design a Hill specific MHRA Survey form that simplifies data collection process with higher level of respondents' involvement.

**Comment (R1)-8: Title of the paper says "Design and Testing of Multi-hazard Rapid assessment questionnaire". However, neither Design part is not discussed in detail nor the testing part is not discussed. It is suggested to include the same for better understanding by the readers.**

As mentioned earlier, we have revised the title as follow: "Design and Application of a Multi Hazard Risk Assessment Survey Questionnaire for the Indian Himalayan Region". The design methodology has been updated in section 3.1, Overall research methodology is updated in section 3.2 and figure 2. Application and discussion of the proposed survey form has been added in section 4.5 and section 5.0 of the manuscript.

**Response to Referee#2**

**Comment (R2)-1: The state of art presented in this part is poor. I believe that authors should report a full state of the art about the risk related to structures and infrastructure and a literature review about the RVS methods.**

[Figure]

**Response:** We appreciate the reviewer's insightful suggestion. Taking this comment into consideration, we have modified the Introduction part in section 1 of the manuscript. However, Literature on RVS has already been mentioned in section 3.3.2. of the manuscript.

**Comment (R2)-2: Please ensure high quality figure 1 to 5**

**Response:** We thank the reviewer's for highlighting this point and accordingly, we have replaced all the figures (identified by referee-Figure 1 to 5) with high quality pixels. We will send the figures in separate files.

**Comment (R2)-3: Table must be reported as a table and not as a figure**

[Figure]

**Response:** We appreciate the reviewer's insightful suggestion; accordingly, we have updated the Table 1.

**Comment (R2)-4: You should better declare the needs of your study, mainly anticipating what will be the advantages of the proposed procedure and with regard to each methodology reported in the table**

[Figure]

**Response:** We have refined the need of the study in section 2.3. However, advantage of the proposed procedure is already mention in section 5.2 of the manuscript.

**Comment (R2)-5: (QuestionPro, n.d.), is not clear. Please revise it.**

[Figure]

**Response:** We have revised the design methodology in Section 3.1 and detailed the overall methodology adopted in section 3.2 and figure 2 of the manuscript for better clarity.

**Comment (R2)-6: A better description of the methodology must be provided. In addition, on what scientific base did authors propose this method?**

[Figure]

**Response:** We greatly appreciate the reviewer's thoughtful suggestion. Taking this comment into consideration, we have revised the methodology in section 3.2 and figure 2. We have also detailed it for better understanding and clarity on the overall methodology adoption.

**Comment (R2)-7: This part (3.3.3) is not clear and it is poor. Please provide a complete definition of the levels. Are the levels reported in the graphical outlines in Fig1**

[Figure]

**Response:** We have revised the section 3.3.3 and incorporated it in the methodology figure 2.

**Comment (R2)-8: All RVS method can be reported above, in a state-of-the-art section, before the methodology presentation.**

[Figure]

**Response:** We appreciate the reviewer's thoughtful suggestion. However, considering the structure of the manuscript, after a thorough discussion, we are continuing the flow of the structure as before, i.e. to combine all information related to literature reviews in section 3.3 of the manuscript, including information about RVS.

**Comment (R2)-9: Where are the results of the pilot survey? Which are the resulting values? This part must be integrated**

[Figure]

**Response:** We greatly appreciate the reviewer's thoughtful suggestion (which will definitely enhance our work) and we agree with it. Taking this comment into consideration, we have added Pilot Survey of 10 schools and its results in section 4.5 and discussion about its result in section 5.3 of the manuscript.

**Comment (R2)-10: What is the main advantage of the proposed procedure? Is there a calibration process? Is there a way to validate the obtained results?**

[Figure]

**Response:** Taking this comment into consideration, we have revised the conclusion part.

We would like to thank the referees once again for taking the time to examine our manuscript. Our manuscript quality has been enhanced by your comments and suggestions.

---

## Referee Report (RR1)

Following are some general/specific observations and comments which needs to be incorporated if agreed:

Line 23-29:
If it has some international application in some other countries, give some names or suggest in brief what modifications can be made to convert them for international use specifically in some Asian countries.

Line 35:
It's researcher's work, so it should be clear what hazards have been considered. Do not use term etc. Name hazards which you are considering in particular.

Line 87-88: give reference of this data

Line 92:

There are two Union Territories not one accordingly modification are needed in text.

Line 96:

How tourism helps in construction of Dam Project. I think something is missing. Please modify.

Line 128:
In table 1 in column 3 Location- what these numbers in column represent clarify.

Table 2 is summary of Table3. This may be stated here. In fact here author may give other such studies if carried out by some states. This part is need assessment only so why Table 2 is required which is summary of your main work.

3. Onwards:

(i) Methodology of designing forms is suggested by Author or adopted from somewhere not clear. If adopted reference is required

(ii) As per figure 2, after MHRA Form application, on 20 schools what modifications have been done in form if any? It may be included.

(iii) No need to discuss four levels assessment when it is not done and used.

(iv) One relevant code to assess seismic vulnerability IS15988 is missing
(v) Please change the  IS 1893-2002  to IS 1893-2016

(vi) Figure 3: It is BIS Map as per IS 1893 not GSI. Figure quality needs improvement.

Line365 onwards:

(i) BMTPC is based on typology /material as per Census of India only this may be included.

(ii) Quality of Table 3 requires improvement.

(iv) Quality of Figure 4 needs improvement It is part of Table 3 only.

3.3.5
Include some details of Industrial hazard also.
4.
(i) Line 448: BIS Map not GSI

(ii)Write details as per  steps of your methodology given in figure 2, in sequence.

4.2.1
(i) Should establish your hard work of going to field in view of your methodology

Some General Observations:

1. If wind hazard is done as per IS code give reference. Similarly for hazards other than seismic give references which you must have referred in case not included.

2. You are considering building related fire vulnerabilities and not forest fire. It may be clarified.

3. Climate change has been addressed in form. It has to be included in text also.

4. It seems you have plan to assess seismic vulnerability of structure and superimpose all vulnerabilities together on regional plan in future if so describe it in end as future research in progress.

5. It will be good to represent designed form in pictorial manner or in brief. Detailed Form can be attached as Annexure appropriately if authors agree.

---

## Editor Decision (ED1)

[revised manuscript text omitted]

---

## Author Response (AR2)

**Response to Referee#1**

**Note: Comment of Referee 1 is in Green and Referee 2 is in blue colour**

We would like to thank both the reviewers for taking the time to assess our manuscript and giving positive and insightful comments on it that has enhanced our manuscript.

We have addressed all the concerns they raised. Our responses to each of their comments are as follows:

*Line 23-29: A comprehensive review of various existing survey forms for Risk assessment has found that the survey questionnaires themselves have not been designed or optimised, specifically, for hill communities. Hill communities are distinctly different from low-land communities, with distinct characteristics and susceptibility to specific hazard and risk scenarios. Previous studies have, on the whole, underrepresented the specific characteristics of hill communities, and the increasing threat of natural disasters in the IHR creates an imperative to design hill-specific questionnaires for multi-hazards risk assessment.*

**Comment (R1)-1: In line 23-29, If it has some international application in some other countries, give some names or suggest in brief what modifications can be made to convert them for international use specifically in some Asian countries.**

**Response:** We appreciate the reviewer, for bringing out attention to this point. We agree with you about its scope in other Asian countries with Himalayas, however our paper has applied only in Indian Himalayan region till now and focus on the same. Taking your suggestion into consideration, addition have been made in the conclusion section.

*Line 35: The survey form covers data related to vulnerability from Earthquake (Rapid Visual Screening), Flood, Landslide, High Wind, Industrial etc.*

**Comment (R1)-2: In line 35, It's researcher's work, so it should be clear what hazards have been considered. Do not use term etc. Name hazards which you are considering in particular.**

**Response:** We appreciate the reviewer for pointing this out. Taking this comment into consideration, we have revised this line.

*Line 87-88: The Indian Himalayan Region (IHR) straddles the northern latitudes of 26 20' and 35 40', and the eastern latitudes of 74 50' and 95 40'.*

**Comment (R1)-3: In line 87-88, give reference of this data**

**Response:** Taking this comment into consideration, we have added Reference to this line.

*Line 92: There are a total of 12 Indian Himalayan states and 1 Union territory as shown in Figure 1, which has 109 administrative districts (Kala, 2014).*

**Comment (R1)-4: In line 92, There are two Union Territories not one accordingly modification are needed in text.**

**Response:** We appreciate the reviewer's insightful observation and taking this comment into consideration, we have updated it in text as well as in figure 1.

[Figure]

**INDIA**

Map Source:
(Siddique et al., 2019)

**Indian Himalayan Region (IHR)**
2 Union Territories and 11 States of India
(Total 13)

**2 Union Territories**
1. Ladakh
2. Jammu & Kashmir

**11 Himalayan States**
3. Himachal Pradesh    9. Assam* (Karbi Anglong & North Cachar)
4. Uttarakhand          10. Meghalaya
5. Sikkim               11. Manipur
6. West Bengal* (Darjeeling)  12. Mizoram
7. Arunachal Pradesh    13. Tripura
8. Nagaland             * Only Hill Districts

Map Source:
(NMHS, n.d.)

*Line 96: Tourism is a lucrative business in IHR (Gaur and Kotru, 2018) and it contributes to support a lot of construction projects like dams across the region (Kala, 2014).*

**Comment (R1)-5: In line 96, How tourism helps in construction of Dam Project. I think something is missing. Please modify.**

**Response:** We appreciate the reviewer's insightful observation. Taking this point into consideration, modification has been done in this line.

*Line 128: Table 1: Major Disaster Events in IHR, Column 3 is Location. Source: adapted from (BMTPC, 2019), (Kumar et al., 2016).*

**Comment (R1)-6: In line 128, In table 1 in column 3 Location- what these numbers in column represent clarify.**

**Response:** Taking this comment into consideration, we have added the description in bracket (at third column of table 1) as Latitude and Longitude.

**Comment (R1)-7: Table 2 is summary of Table3. This may be stated here. In fact, here author may give other such studies if carried out by some states. This part is need assessment only so why Table 2 is required which is summary of your main work.**

**Response:** We appreciate the reviewer's insightful recommendation. I agree with you that Table 2 is the summary, however it is to show the need of this study in a summarized way. On the other hand, Table 3 is the detailed comparison of all the existing survey form used in India and the proposed survey form. Because of this reason, we would like to keep it as it is.

**Comment (R1)-8: Section 3. Onwards:**

**Comment (R1)-8.1: (i) Methodology of designing forms is suggested by Author or adopted from somewhere not clear. If adopted reference is required**

**Response:** Taking this point into consideration and for better clarity, we have updated the caption of figure 2 as "Methodology adopted by author"

**Comment (R1)-8.2: (ii) As per figure 2, after MHRA Form application, on 20 schools what modifications have been done in form if any? It may be included.**

**Response:** We appreciate the reviewer's insightful suggestion. I would like to highlight that application on 10 schools (please note its 10 schools) was done with the final proposed survey form. Before application in these schools, the gaps identified in the existing survey forms and observation during preliminary survey are described in section 3.4.1 and 4.2.1 respectively.

**Comment (R1)-8.3: (iii) No need to discuss four levels assessment when it is not done and used.**

**Response:** We appreciate the reviewer's insightful suggestion. We agree with the reviews, removing this section will not impact the literature of the manuscript. Thus we removed this part as per your suggestion.

**Comment (R1)-8.4: (iv) One relevant code to assess seismic vulnerability IS15988 is missing**

**Response:** Taking this point into consideration, we have added IS-15988 (2013) in section 4.4.

**Comment (R1)-8.5: (v) Please change the IS 1893-2002 to IS 1893-2016**

**Response:** We appreciate the reviewer's insightful suggestion. I would like to highlight that IS-1893-2016 is already mentioned in section 4.4. However, taking this point into consideration, we have removed IS-1893-2002.

**Comment (R1)-8.6: (vi) Figure 3: It is BIS Map as per IS 1893 not GSI. Figure quality needs improvement.**

**Response:** We appreciate the reviewer's insightful suggestion. Taking this point into consideration, we have added all the sources including BIS. All the high quality images has already been send separately.

**Comment (R1)-9: Line365 onwards:**
**Comment (R1)-9.1: BMTPC is based on typology /material as per Census of India only this may be included.**

**Response:** Taking this point into consideration, we have updated it in section 3.3.3.7.

**Comment (R1)-9.2: Quality of Table 3 requires improvement.**

**Response:** We thank the reviewer's for highlighting this point. This is to inform you that a file with high pixel quality is already send as a separate file.

**Comment (R1)-9.3: Quality of Figure 4 needs improvement It is part of Table 3 only.**

**Response:** We thank the reviewer's for highlighting this point. This is to inform you that a file with high pixel quality is already send as a separate file.

**Comment (R1)-10: In Section 3.3.5, Include some details of Industrial hazard also.**

**Response:** We thank the reviewers for this suggestion. As existing multi-hazard survey forms used in India does not focus on Industrial hazard (table 2), we have not mentioned it in section 3.3.3 and 3.3.4. However, as a value addition of the proposed survey form, we have already mentioned it in table 4 and section 5.2.

**Comment (R1)-11: In Section 4.**

*Line 448 onwards: Whilst the Himalayan region is prone to earthquakes as per India's Seismic Zonation Map (Figure 3) prepared by the Geographical Survey of India (GSI), the proposed survey form also covers other hazards like landslide, flood, industrial explosion/emissions, fire, hydro-climatic factors, etc., which will be addressed one by one in this paper.*

**Comment (R1)-11.1: (i) Line 448: BIS Map not GSI**

**Response:** We appreciate the reviewer's insightful recommendation. Taking this point into consideration we have modified it.

**Comment (R1)-11.2: (ii) Write details as per steps of your methodology given in figure 2, in sequence.**

**Response:** We appreciate the reviewer's insightful suggestion. I would like to highlight that the methodology is already explained in section 3.2. and detail of every step is explained in section 4.

**Comment (R1)-12: In Section 4.2.1**
**Should establish your hard work of going to field in view of your methodology**

**Response:** We thank you for appreciating our hard work.

**Comment (R1)-13: Some General Observations:**
**Comment (R1)-13.1: If wind hazard is done as per IS code give reference. Similarly for hazards other than seismic give references which you must have referred in case not included.**

**Response:** Considering this point we have added all the references in section 4.4

**Comment (R1)-13.2: You are considering building related fire vulnerabilities and not forest fire. It may be clarified.**

**Response:** We appreciate the reviewer's insightful opinion from readers point of views. Taking this point into consideration, we have identified all related words and modified it.

**Comment (R1)-13.3: Climate change has been addressed in form. It has to be included in text also.**

**Response:** We appreciate the reviewer's suggestion. Taking this point into consideration, we have made the addition in section 5.2.

**Comment (R1)-13.4: It seems you have plan to assess seismic vulnerability of structure and superimpose all vulnerabilities together on regional plan in future if so, describe it in end as future research in progress.**

**Response:** We appreciate the reviewer's futuristic suggestions. In order to strengthen it further, we have made the modifications in the conclusion section.

**Comment (R1)-13.5: It will be good to represent designed form in pictorial manner or in brief. Detailed Form can be attached as Annexure appropriately if authors agree.**

**Response:** We appreciate the reviewer's helpful proposal from surveyor's point of views. We can provide a separate annexure as "Survey form", if required.

**Response to Referee#2**

We would like to thank the reviewers for taking the time to assess our manuscript and giving positive and insightful comments on it that has enhanced our manuscript.

We have addressed all the concerns they raised. Our responses to each of their comments are as follows:

**Comment (R2)-1: The manuscript contents useful information regarding the multi-hazard risk assessment survey in the Indian context. However, due to rather scanty write-up of the manuscript, the idea, work conducted, and outcomes of the presented research are not conveyed well.**

**Response:** We would like to thank the reviewer for their comments. We have considered all your insightful suggestion and provided our comments as follow.

**Comment (R2)-2: The National Building Code of India (2016) published by the Bureau of Indian Standards (BIS), New Delhi, describes the 'Multi-Hazard Risk Concept' and 'Multi-Hazard Prone Area,' which can be helpful for the readers if included in the review presented by the authors in the manuscript.**

**Response:** Thank you so much reviewer for sharing this information. We agree and have updated it in the introduction section.

**Comment (R2)-3: (1) The need and relevance of variety of multi-hazard scenarios and risk assessment thereof for infrastructure in case of India should be highlighted in the introduction. (2) This aspect in the context of, multi-hazard analysis and design guidelines: recommendations for structure and infrastructure systems in the Indian context, which has been discussed earlier-on provides basis for further investigation, as being dealt with in the present study. (3) Moreover, the existing strategies of risk assessment of the infrastructures in areas of India subjected to multi-hazard should be mentioned to provide a broad perspective in this research area.**

**Response:** We appreciate the reviewer's suggestion. (1) We agree with the reviewer however the brief about the variety of multi-hazard scenarios and risk assessment used in India is already explained in section 3.3.3 and need of it is explained in section 2.3 of the manuscript. (2) Study of design guidelines and recommendations from Indian context is in itself a huge study. We have planned it to be the part of separate article, as this manuscript focus only on design and application of the proposed survey form. (3) The brief about existing strategies of risk assessment used in India has been already explain n in section 3.3.3.

**Comment (R2)-4: Also, the authors mainly talked about multi-hazard risk assessment in the Indian context; however, the study of how the assessment is being otherwise done globally, and associated future challenges can certainly improve the questionnaire, i.e., learning lessons from other parts. Hence, multi-hazard analysis and design of structures: status and research trends, which should provide that kind of global perspective to the readership should be included in the present manuscript.**

**Response:** We agree with the reviewer that further elaborating lesson learnings from other part of the world could provide global perspective to the readers. However, we believe that the Multi-Hazard Risk Assessment executed globally is in itself a huge study (brief of some of it is describe in section 3.3.4 of the manuscript) and our manuscript focuses only on the Indian Himalayan Region; thus, we would like proceed as it is.

**Comment (R2)-5: The manuscript has mainly focused on the natural hazards in the manuscript. However, multi-hazards can include manmade (/accidental) hazards, such as blast, explosion, fire outbreak, etc. which can cause extensive risk to communities and infrastructures, even in the area of question in the present manuscript. Therefore, a section needs to be added to the questionnaire regarding such manmade hazards.**
**Response:** We appreciate the reviewer's insightful suggestion. We would like to highlight here that the proposed survey form consists of questions related to man-made hazards like Industrial hazard (Part B, question no. 17 to 20), Fire vulnerability (Part B, question no. 21 to 35) and non-structural risk (Part B, question no. 40 to 42). However other man-made hazards are beyond the scope of this research.

**Comment (R2)-6: The "mixed use", terminology in the survey form can be replaced with "combined use" for better understanding. There are many sections in the survey form, where the asked information may not be clear, especially for low-skilled engineers or surveyors; therefore, this reviewer recommends providing symbolic figures for better understanding. For instance, Sections 57 and 64 (grade of cracks) can be presented with an illustrative figure. Furthermore, another column can be added in many sections in case of not finding any existing option true; for instance, in Sections 58 and 65, other failures and cracks can be added.**

**Response:** We appreciate the reviewer's insightful observations and suggestion. Taking this point into consideration, we have added symbolic figure in the options of Question 58 – type of wall failures. However, on the other hand, as per URDPFI Guidelines, the terminology used for such Landuse of the building is "Mixed use", thus we would like to remain with the same terminology and this is to highlight that the pictorial explanation of cracks is already given in question 63 and 64 of the proposed survey form (Part-A), however, question 57 does not required pictorial explanation.

**Comment (R2)-7: (1) In the section "Pounding Effects Details "in the survey form, instead of the subsection "Quality of adjacent building" can be replaced with material and number of floors of adjacent building for checking the capacity of the adjacent building. Subsection 79 can appear after the point mentioned in Subsection 80. (2) The definition of short and long columns provided in the survey form needs to be improved for clear understanding. (3) In Table 6a, providing the options of the range of wind speeds can be more helpful than the option of average wind speed. (4) The authors have conducted a pilot survey; however, all necessary details of the site conditions and pictures need to be included.**

**Response:** We appreciate the reviewer's insightful observations and suggestion. Taking this comment into consideration, (1) in section "Pounding Effect details", we have switched the question 79 with question 80 in the proposed survey form. However, quality of adjacent building has already been asked in question 81 with 5 ranges of quality from very good to very bad. However separate analysis of individual adjacent building will be more beneficial. (2) The explanation of short and long column has already been explained in the pictorial manner wherever required, example question 98, 107, 111 of the proposed survey form. (3) Range of wind speed is modified in the proposed survey form. (4) As this manuscript focus on the designing and application of the survey form, the detail of pilot survey is in itself a huge part and can be a separate article. However, taking this point into consideration, we have added the pictures of 10 school buildings in section 4.5.2. In past 1 years, the proposed survey form is applied over 500+ buildings and we are working for the next article on it.

**Comment (R2)-8: In addition, there are several editorial mistakes throughout the manuscript, which can and must be corrected. Some of them are mentioned in Points 7 and 8 merely for indicative reference purposes. There is a repetition of the same content and sentences, which can be avoided for better readability. The complete form of abbreviation used, such as MS/SS, should be mentioned.**

**Response:** We appreciate the reviewer's insightful suggestion. In order to strengthen it further, we have done all the modifications including providing the complete full form of the abbreviations in the proposed form after every section where applicable.

**Comment (R2)-9: (1) The whole manuscript has inconsistencies in formatting, font size, and unnecessary capitalization of the first letter of the words. While citing the tables and figures in the text, the first letter should be capitalized; for example, in line 170, it can be "Table 2". (2) Also, the quality of the figures and sub-figures require enhancement. (3) The references should be arranged with a consistent formatting style.**

**Response:** We appreciate the reviewer's helpful suggestion. Taking this comment into consideration, (1) We have identified the text (table and figure) and capitalized the first letter as Table and Figure (2) The images with better high quality pixel size is already send as a separate file (3) Referencing style is improved as per journal's guidelines.

**With the suggestions provided, quality of the manuscript can be improved significantly, and the manuscript can be beneficial for engineers and surveyors in assessing the multi-hazard risk for communities and infrastructures.**

**Response:** Thank you so much for your appreciation and comments.

---

## Author Response (AR3)

**Response**

**Part I: Additional private note**

We are very grateful to all the reviewers and editorial team for your comments and your time for reviewing our manuscript. All the grammatical mistakes, abbreviation etc. have been modified as per your comments. Some response to your comments are as follows:

Line 77
**Comment 1:** MHRA has not been defined
**Response:** Full form of MHRA is added

Line 145
**Comment 2:** Are you missing the name of the river here? Or its in general
**Response:** Thank you for pointing it out. The line has been modified for better clarity.

Line 190
**Comment 3:** "and many more" written is unclear
**Response:** Modification is done.

Line 205-206
**Comment 4:** This is not clear; the length of the questionaries should be dictated by the need to cover all essential parts? Or something else should be dictate the length?
**Response:** Thank you for highlighting it. Modification has been done for more clarity.

Line 205-214
**Comment 5:** It is very complicated to put all of this in one sentence. I would suggest to structure it like (1) … (2)..
**Response:** Thank you for your suggestion. I appreciate it. Based on your suggestion, modification is done.

Line 289, 322, 332, 339, 377 and 383
**Comment 6:** For all of these, the format should be: "refer to Arya 2006 for …, this applies also to line 289, 322, 332, 339, 377 and 383
**Response:** Thank you so much for this insightful suggestion. The mentioned lines are modified.

Line 329
**Comment 7:** Citation should be given here.
**Response:** Thank you for pointing it out. Citation added as per the suggestion

Line 335
**Comment 8:** There is no need to state the title of the paper, just give the citation. "is shown in kumar et at (2016)
**Response:** Modification is done as per the suggestion.

Line 342
**Comment 9:** This should give the citation
**Response:** Citation added

Line 409-410
**Comment 10:** SVA and DVA have not been defined
**Response:** Thank you for highlighting it, as per suggestion, the full form of SVA and DVA is added.

Line 571

**Comment 11:** It is not clear what "other" refers to here. You can either leave it out or say the proposed or new multi hazard survey.

**Response:** Thank you for pointing it out. The Modification is done by removing "other".

Line 598-599

**Comment 12:** these don't need to be capitalized.

**Response:** the modification is done as per your suggestion.

Line 632

**Comment 13:** this has not been defined

**Response:** Full form NSRA is added as per your suggestion.

Line 643

**Comment 14:** I think that many people won't know the word "almirah".

**Response:** Almirah is changed with "cabinets"

Line 696

**Comment 15:** @Research is being undertaken@ sounds like ongoing work to further develop the survey form but it seems that here you are describing the results of this study. If that is the case, then it is repetitive, and I would suggest to remove this sentence.

**Response:** I agree with you. As per your suggestion, I removed that sentence.

I would like to thank you for correcting all the grammatical mistakes. All the identified grammatical mistakes are corrected in the submitted revised manuscript (march 2023)

**Part II: Notification to the authors during file validation**

1.Coloured or marked text in *.pdf manuscript file is not allowed. Please provide a clean version of *pdf manuscript file (with black text) with the next revision.

**Response:** Thank you so much for the information. Black text version is submitted in this revised manuscript submission.

2. Please note that single tables` panels with their own captions are not possible in the final version. Please adapt them according to our guidelines:https://www.natural-hazards-and-earth-system-sciences.net/submission.html (section "Figures & Tables").

**Response:** I have gone to the guidelines provided. I have updated the figure 3 by removing caption from the image.

3. Regarding the figure 7: for the next revision, please check if your figures containing photos require a copyright statement/image credit and add it to the figures (or captions) (https://publications.copernicus.org/for_authors/manuscript_preparation.html#figurestables -> Reproduction and reuse of figures and tables). If these figures were entirely created by the authors, there is no need to add a copyright statement or credit. In that case it is important that you confirm this explicitly by email.

**Response:** Figure 7 are the photos clicked by author 1

4. Your tables contain coloured cells or/and coloured values. Please note that this will not be possible in the final revised version of the paper due to HTML conversion of the paper. When revising the final

version, you can use footnotes or italic/bold font.

**Response:** Thank you for the information.

5. Supplement has been removed, as it did not contain additional materials, but figures from the manuscript. Please note that you will be asked to provide these materials at other stages of the review process.

**Response:** Thank you for the information. Yes, I will provide these materials whenever asked.

---

## Author Response (AR4)

**RESPONSE**

**Comment 1**. Please note that single tables` panels with their own captions are not possible in the final version. Please adapt them according to our guidelines:https://www.natural-hazards-and-earth-system-sciences.net/submission.html (section "Figures & Tables").

**Responses:** Thank you so much for your consideration in making our manuscript best. Taking this comment into consideration I have made two chances as follows:

- Line 182: Table 2 is updated by removing captions from the table
- Line 357: Table 3 is updated by removing captions from the table, and added captions on the title of the table 3

**Comment 2.** Your tables contain coloured cells or/and coloured values. Please note that this will not be possible in the final revised version of the paper due to HTML conversion of the paper. When revising the final version, you can use footnotes or italic/bold font.

**Responses:** Thank you so much for highlight this point. Taking this comment into consideration I have made three chances as follows:

- Line 559: Table 7 is updated by removing colour code row and I have converted this table into tabular format (which was earlier in image format)
- Line 570:Table 8 is updated by removing colours from row 1 (Risk status).
- Line 577: Table 9 is updated by removing colours from row 1 (risk status) and I have converted this table into tabular format (which was earlier in image format). As I converted this figure (line 577) into table. It become table 9 and Figure 7 (line 581) becomes figure 6.

**Comment 3.** Supplement has been removed, as it did not contain additional materials, but figures from the manuscript. Please note that you will be asked to provide these materials at other stages of the review process.

**Responses:** I would like to highlight here that. Taking this comment into consideration I have provided the following as supplement, please feel free to format it as per the requirement of the journal

- Line 496 to 542: Table 5 and 6 are proposed survey form. I have provided the excel sheet of the complete survey form in the supplement during this submission